# Redox regulation of K$_V$7 channels through EF3 hand of calmodulin

Eider Nuñez[1], Frederick Jones[2], Arantza Muguruza-Montero[1], Janire Urrutia[1], Alejandra Aguado[1], Covadonga Malo[1], Ganeko Bernardo-Seisdedos[3], Carmen Domene[4,5], Oscar Millet[6], Nikita Gamper[2], Alvaro Villarroel[1]*

[1]Instituto Biofisika, CSIC-UPV/EHU, Leioa, Spain; [2]School of Biomedical Sciences, Faculty of Biological Sciences, University of Leeds, Leeds, United Kingdom; [3]Atlas Molecular Pharma S.L, Derio, Spain; [4]Department of Chemistry, University of Bath, Bath, United Kingdom; [5]Department of Chemistry, University of Oxford, Oxford, United Kingdom; [6]Protein Stability and Inherited Disease Laboratory, CIC bioGUNE, Derio, Spain

**Abstract** Neuronal K$_V$7 channels, important regulators of cell excitability, are among the most sensitive proteins to reactive oxygen species. The S2S3 linker of the voltage sensor was reported as a site-mediating redox modulation of the channels. Recent structural insights reveal potential interactions between this linker and the Ca$^{2+}$-binding loop of the third EF-hand of calmodulin (CaM), which embraces an antiparallel fork formed by the C-terminal helices A and B, constituting the calcium responsive domain (CRD). We found that precluding Ca$^{2+}$ binding to the EF3 hand, but not to EF1, EF2, or EF4 hands, abolishes oxidation-induced enhancement of K$_V$7.4 currents. Monitoring FRET (Fluorescence Resonance Energy Transfer) between helices A and B using purified CRDs tagged with fluorescent proteins, we observed that S2S3 peptides cause a reversal of the signal in the presence of Ca$^{2+}$ but have no effect in the absence of this cation or if the peptide is oxidized. The capacity of loading EF3 with Ca$^{2+}$ is essential for this reversal of the FRET signal, whereas the consequences of obliterating Ca$^{2+}$ binding to EF1, EF2, or EF4 are negligible. Furthermore, we show that EF3 is critical for translating Ca$^{2+}$ signals to reorient the AB fork. Our data are consistent with the proposal that oxidation of cysteine residues in the S2S3 loop relieves K$_V$7 channels from a constitutive inhibition imposed by interactions between the EF3 hand of CaM which is crucial for this signaling.

**\*For correspondence:**
alvaro.villarroel@csic.es

**Competing interest:** The authors declare that no competing interests exist.

## Editor's evaluation

This useful study provides insights into mechanisms underlying oxidation regulation of Kv7 channels that contributes to regulating neuronal excitability. The experimental evidence in support of the major claims is solid, although it could be improved upon by studies on the holo-channel. The work will be of general interest to ion channel biophysicists and cell biologists.

## Introduction

The generation of abnormally high levels of reactive oxygen species (ROS) is linked to cellular dysfunction, including neuronal toxicity and neurodegeneration (*Abdullaeva et al., 2022*; *Sahoo et al., 2014*; *Sohal and Orr, 2012*). In addition, ROS are important mediators of normal cellular functions in multiple intracellular signal transduction pathways (*Dantzler et al., 2019*; *Miki and Funato, 2012*; *Wani and Murray, 2017*; *Weidinger and Kozlov, 2015*). ROS generation induces oxidative modifications and augmentation of M-currents in neurons, which provides protective effects on oxidative stress-related

neurodegeneration (*Bierbower et al., 2015*; *Gamper et al., 2006*; *Vigil et al., 2020*). $K_V7$ channels, the substrate of the $K_V7$-mediated M-current, are among the most sensitive proteins that respond to ROS production (*Gamper et al., 2006*; *Jones et al., 2021*; *Sahoo et al., 2014*).

Superoxide anion radicals ($O_2\bullet-$), hydroxyl radicals ($\bullet OH$), peroxynitrite ($ONOO-$), and hydrogen peroxide ($H_2O_2$) are the main ROS produced in cells (*Mittal et al., 2014*). These molecules display different reactivity, concentration and lifetime, and most probably play different roles in signal transduction and oxidative stress. Oxidation of cysteine thiol side chains mediated by $H_2O_2$ is the most recognized and studied redox reversible post-translational modification. Because of its relative stability and ability to cross the plasma membrane, $H_2O_2$ has been shown to be important in a variety of neurophysiological processes, including neurotransmission, ion channel function, and neuronal activity (*Gamper and Ooi, 2015*; *Kamsler and Segal, 2007*; *Lee et al., 2015*; *Patel and Rice, 2012*).

Augmentation of the M-current can be induced by an external $H_2O_2$ concentration as low as 5 µM (*Gamper et al., 2006*) or even in the nM range (*Abdullaeva et al., 2022*). The M-current flow through channels formed of neuronal $K_V7$ subunits ($K_V7.2$-$K_V7.5$, encoded by KCNQ2-5 genes). These tetrameric channels open at the subthreshold membrane potentials and dampen cellular excitability (*Adams, 2016*; *Soldovieri et al., 2011*). $K_V7$ channels have a core architecture similar to other voltage-dependent potassium channels (*Li et al., 2021a*; *Li et al., 2021b*; ; *Sun and MacKinnon, 2017*; *Sun and MacKinnon, 2020*; *Xu et al., 2013*): they have six helical transmembrane domains (S1–S6) with the voltage sensor formed by S1–S4, followed by a pore domain (S5–S6), which continues into a cytosolic C-terminal region. The C-terminus of $K_V7$ channels contains five helical regions: helices A–D and TW helix between hA and hB. The latter region forms the calcium responsive domain (CRD) with helices AB adopting an antiparallel fork disposition (*Sachyani et al., 2014*). Four C-helices from each subunit come together to form a stem perpendicular to the membrane. This stem continues with an unstructured linker that connects to helix D, which forms a tetrameric coiled-coil structure that confers subunit specificity during subunit assembly (*Sachyani et al., 2014*).

All $K_V7$ channels require the association of calmodulin (CaM) to the CRD to be functional (*Gamper and Shapiro, 2003*; *Wen and Levitan, 2002*; *Yus-Najera et al., 2002*). Helices AB are embraced by CaM forming a compact structure just under the membrane that can move as a rigid body, with a region connecting S6 and helix A acting as a hinge (*Li et al., 2021a*; *Li et al., 2021b*; *Xu et al., 2013*). CaM is the main adaptor protein that confers $Ca^{2+}$ sensitivity to an ample array of eukaryotic proteins and is composed of two highly homologous lobes joined by a flexible linker. In solution, each lobe operates almost independently of the other and contains two similar $Ca^{2+}$-binding EF-hands (*Sorensen and Shea, 1998*). This distinct signaling mediated by each CaM lobe was revealed early in *Paramecium* when it was discovered that mutations at the N-lobe affected a $Ca^{2+}$ operated $Na^+$ conductance, whereas mutations at the C-lobe affected a $Ca^{2+}$ dependent $K^+$ conductance (*Kung et al., 1992*).

A structure of the non-neuronal $K_V7.1$ subunit trapped in a non-functional conformation with the voltage-sensor disengaged from the pore suggests that the EF3-hand of CaM may interact with the voltage sensor (*Kang et al., 2020*; *Li et al., 2021b*; *Sun and MacKinnon, 2017*) at a site essential for M-current redox modulation (*Abdullaeva et al., 2022*; *Gamper et al., 2006*). This 3D configuration has been assumed to confer a preferential use of EF3 during signaling on $K_V7$ channels (*Chang et al., 2018*; *Kang et al., 2020*; *Tobelaim et al., 2017*; *Zhuang and Yan, 2020*).

Here, we address the role of CaM on redox modulation of $K_V7$ channels, finding a critical role of EF3-hand. We show that $H_2O_2$ interrupts crosstalk between the S2S3 linker and the EF3-hand of CaM in a $Ca^{2+}$-dependent manner. We have monitored the role of each EF-hand in the gating process of the CRD, characterized by the opening of the AB fork (*Bernardo-Seisdedos et al., 2018*). By studying purified signaling components in a well-controlled in vitro setting, we avoided unwanted off-target effects arising from interactions with other domains of the channel; we then tested our structural findings in cellulo. The prevailing view is that there is a high degree of cooperativity between the pair of EF-hands within each lobe, such that each lobe operates as a unit regarding $Ca^{2+}$ binding and signaling and that the 3D arrangement is required for EF-hand specific allosteric signaling of $K_V7$ channels (*Kang et al., 2020*; *Zhuang and Yan, 2020*). In contrast, we show an additional level of specialization whereby just one EF-hand is critical for $Ca^{2+}$ signaling, and the direction of gating can change upon interaction between S2S3 loop domain and solvent-exposed EF-hands. Thus, preferential signaling through EF3 is an intrinsic property not derived from the 3D arrangement revealed by the available

$K_V7$ structures. The emergence of this novel mode of CaM modulation promises generalization to complexes with the EF-hands interacting with solvent-exposed regions of the target proteins.

## Results

### CaM plays a critical role in $H_2O_2$-mediated regulation of $K_V7.4$ channels

We have previously shown that cysteine residues present in the unusual long linker between S2S3 transmembrane segments of $K_V7$ channels are critical for $H_2O_2$-dependent potentiation (*Gamper et al., 2006*). Recent studies suggest structural and functional interactions between this loop and CaM (*Chang et al., 2018*; *Kang et al., 2020*; *Sun and MacKinnon, 2017*; *Tobelaim et al., 2017*; *Zhuang and Yan, 2020*). To test a possible role of CaM in redox modulation, we used the perforated patch clamp method to measure $K_V7.4$ activity in response to $H_2O_2$. Human KCNQ4 cDNA was co-expressed in HEK293 cells with either CaM or CaM mutants that, by replacing the aspartate residue with alanine in

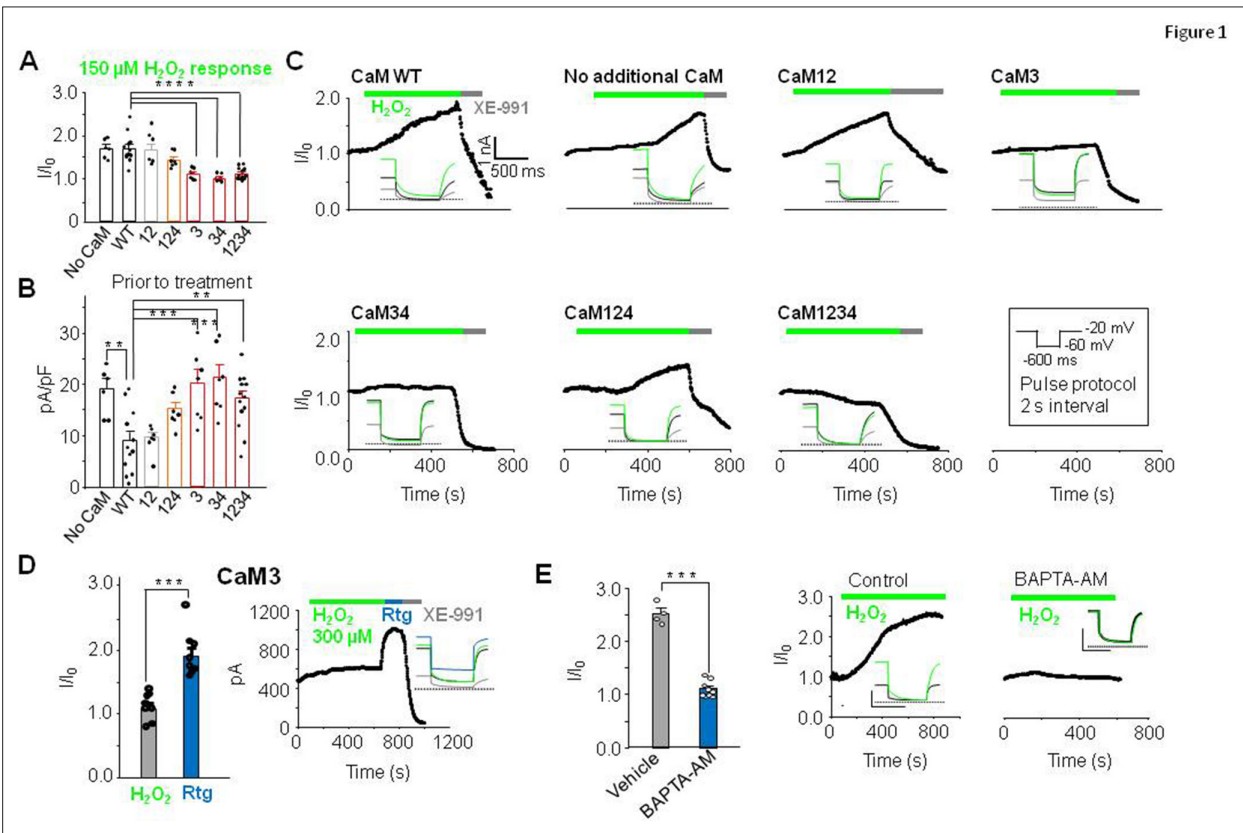

**Figure 1.** EF3 hand $Ca^{2+}$ binding capacity of CaM is required for $H_2O_2$-mediated potentiation of $K_V7.4$. (**A**) Response of $K_V7.4$ transfected HEK293 cells to 150 µM $H_2O_2$ (normalized steady-state current at –60 mV, $I/I_0$) when transfected with wild-type CaM (CaMWT n=12), mutant CaMs lacking $Ca^{2+}$ binding to one or more EF hands. The number in X-axis of panel B applies and pertains to the EF hand unable to bind $Ca^{2+}$ (CaM12 n=7, CaM124 n=7, CaM3 n=7, CaM34 n=7, and CaM1234 n=13) or with no additional CaM transfected (No CaM, n=6). (**B**) Current density (pA/pF; –60 mV) of $K_V7.4$ transfected cells prior to treatment with $H_2O_2$. (**C**) Representative currents at –60 mV in response to 150 µM $H_2O_2$ followed by 10 µM XE-991. Inset: representative current traces from each condition. (**D**) Comparative response of cells transfected with $K_V7.4$ and CaM3 to 300 µM $H_2O_2$ and 10 µM retigabine (n=8). (**E**) $Ca^{2+}$ dependence of $H_2O_2$ response in cells transfected with $K_V7.4$ and CaMWT. Comparison of 300 µM $H_2O_2$ response in normal or low $Ca^{2+}$ conditions induced by pre-incubation of cells in 10 µM BAPTA-AM for 30 min to chelate intracellular $Ca^{2+}$. Control n=4, BAPTA-AM n=9. Data presented are mean ± SEM, statistical evaluation by independent measures ANOVA with Dunnett's post hoc, **p<0.01, ***p<0.001, and ****p<0.0001 (A and B). A paired (**D**) or unpaired (**E**) two-tailed T test ***p<0.001 and ****p<0.0001.

The online version of this article includes the following source data and figure supplement(s) for figure 1:

**Source data 1.** The current voltage relationship of cells transfected with mutant CaM does not differ significantly from CaMWT.

**Figure supplement 1.** The current voltage relationship of cells transfected with mutant CaM does not differ significantly from CaMWT.

**Figure supplement 2.** Further support for the link between $H_2O_2$, CaM, and the S2S3 linker in $K_V7$ channels.

**Figure supplement 2—source data 1.** Link between $H_2O_2$, CaM and the S2S3 linker in $K_V7$ channels.

the first position of the EF-hands, the $Ca^{2+}$ binding ability of the N-lobe (CaM12), the C-lobe (CaM34), or both (CaM1234; *Geiser et al., 1991*; *Keen et al., 1999*) is disabled (*Figure 1*). Cells were held at –20 mV, and 600 ms voltage pulses to –60 mV were applied every 2 s; $K_V7.4$ activity was monitored as the outward steady-state current amplitude at –60 mV (*Figure 1C*). Bath-application of 150 µM $H_2O_2$ induced a clear augmentation of steady-state currents in the presence of CaM or CaM12 (*Figure 1A–C*, *Figure 1—figure supplement 1*). In contrast, the response was attenuated or precluded in the presence of CaM1234 or CaM34 (*Figure 1A–C*). Because structural and functional studies suggested a critical role of EF3 (*Chang et al., 2018*; *Kang et al., 2020*; *Sun and MacKinnon, 2017*; *Tobelaim et al., 2017*; *Zhuang and Yan, 2020*), we tested the effect of CaM3 and CaM124. Whereas the $H_2O_2$ response in the presence of CaM124 (*Figure 1A–C*) was maintained, it was diminished with CaM3 (*Figure 1D*). Importantly, while the response to $H_2O_2$ was abolished in the presence of CaM3, another $K_V7$ activator, retigabine, still produced strong activation of $K_V7.4$ current under these conditions (*Figure 1D*). Retigabine activates $K_V7$ channels by binding to a hydrophobic pocket between S4 and S5 domains, a site that does not overlap with CaM binding site (*Wuttke et al., 2005*). These results suggest that EF3 of CaM is necessary for augmentation of $K_V7$ channels by $H_2O_2$ specifically.

EF-hand mutations used above mimic $Ca^{2+}$-free (apo) state of the CaM, with CaM1234 being completely $Ca^{2+}$-free, while other mutants are partially $Ca^{2+}$-free. Since CaM1234 prevented the $K_V7.4$ current augmentation by $H_2O_2$ (as did the other mutants containing EF3 mutation), we therefore tested if 'sponging' intracellular $Ca^{2+}$ by pre-incubating the cells with BAPTA-AM also prevent the $H_2O_2$ effect on $K_V7.4$. BAPTA-AM crosses the membrane and release the strong $Ca^{2+}$ chelator BAPTA intracellularly, thereby lowering resting $Ca^{2+}$ levels. The response to oxidation was indeed virtually abolished under these conditions.

As expected, the effect of $H_2O_2$ was absent after substituting the redox-sensitive triplet of cysteine residues at the positions 156, 157, and 158 in the S2S3 linker of $K_V7.4$ by alanine residues (*Figure 1—figure supplement 2*). There was no difference whether WT CaM or CaM1234 was present; in either case, the current produced by CCCAAA $K_V7.4$ was only marginally affected by 150 µM $H_2O_2$ (*Figure 1—figure supplement 2*). Interestingly, all CaM mutants containing EF3 mutations (CaM3, CaM34, and CaM1234) produced small negative shift in $K_V7.4$ voltage dependence (*Figure 1—figure supplement 1*), a finding consistent with a presumed removal of a tonic inhibitory effect of calcified EF3.

Overall, these experiments suggested that EF3 of CaM and cysteine residues in the S2S3 of $K_V7.4$ are necessary for current activation by $H_2O_2$. We hypothesize that binding of $Ca^{2+}$ to EF3 partially inhibits $K_V7.4$; preventing binding or removing $Ca^{2+}$ from this location disinhibits the channel. We further hypothesize that oxidative modification of S2S3 cysteine residues antagonizes the EF3/$Ca^{2+}$ inhibition of $K_V7.4$. Yet, the interpretation of this effect requires caution since CaM over-expression also affects the number of the channels at the plasma membrane (*Etxeberria et al., 2008*; *Gomis-Perez et al., 2017*). To get further insights, we analyzed the behavior of the isolated CRD, without constrains imposed by other channel domains, the membrane, or the complexity of potential intracellular signaling cascades evoked in vivo.

## CaM remains attached to the AB fork

To test the stability of CaM engagement in the CRD, a fluorescent tag was placed in CaM (CaM-YFP) and another in the $K_V7.2$ AB fork (mTFP1-AB). Two complementary assays were performed. In the first one, the complex was established with both components carrying a fluorescent tag, yielding an initial significant FRET value. Then, the complex was incubated up to 3 hr with excess of CaM (10-fold), devoid of any tag. Here, exchange between anchored and free CaM should be accompanied by a reduction in FRET. In the second assay, the complex was established between a tagged AB fork and label-free CaM. Subsequently, the complex was incubated with fluorescently tagged CaM. Exchange of anchored and free CaM should be accompanied by an increase in FRET. No changes in FRET were observed after 3 hr, neither in the presence nor in the absence of free $Ca^{2+}$ (n=4; *Figure 2—figure supplement 1*). Thus, under these in vitro conditions, CaM remained firmly attached to the AB fork. Incidentally, the magnitude of FRET changes between mTFP1-AB, tagged just before helix A, in complex with CaM-YFP, tagged at the C-lobe, are best described as the C-lobe remaining bound to helix A in the presence of $Ca^{2+}$, as previously revealed by NMR (Nuclear Magnetic Resonance) analysis of the complex (*Bernardo-Seisdedos et al., 2018*).

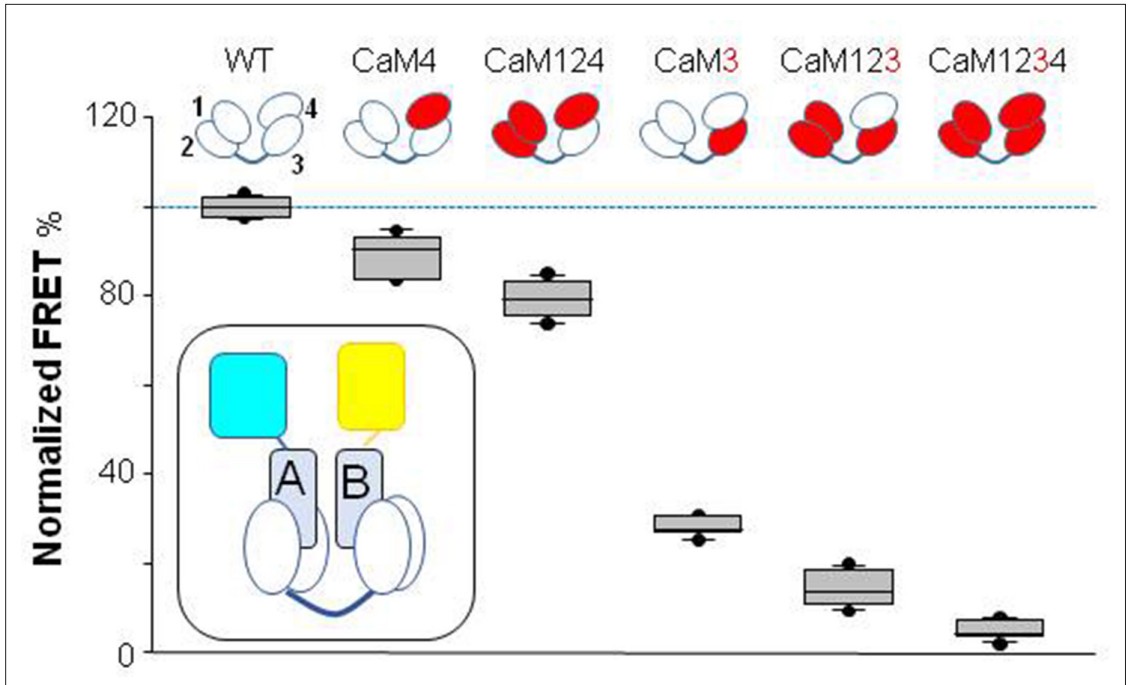

**Figure 2.** Influence of Ca$^{2+}$-binding abolishing mutations in EF hands on Ca$^{2+}$-dependent FRET changes. Top: cartoon representation of CaM mutants. The EF-hands carrying a mutation that preclude Ca$^{2+}$ binding are colored in red. Bottom: box-plot of the relative FRET index change produced by Ca$^{2+}$ for the AB fork in complex with the indicated mutated CaM. Note that in the complex with CaM3 and CaM123, the changes prompted by 16 µM Ca$^{2+}$ were almost obliterated, whereas in the complex with CaM124 the response was preserved. Each plot represents the average of six independent experiments. FRET index was defined as the ratio of the fluorescence peak between mcpVenus (yellow acceptor) and mTFP1 (blue donor). The index was normalized to the value obtained with WT CaM. Experiments were performed at 500 nM of *h*AB:CaM purified complex, in a 1:1 ratio. Inset: cartoon representing the FRET sensor in complex with CaM (mTFP1-hA-hB-Venus/CaM).

The online version of this article includes the following source data and figure supplement(s) for figure 2:

**Source data 1.** Tabulated FRET values for each condition.

**Figure supplement 1.** Residence of CaM in the calcium responsive domain (CRD) complex.

**Figure supplement 1—source data 1.** Residence of CaM in the CRD complex.

**Figure supplement 2.** Box plot of the influence of Ca$^{2+}$-binding abolishing mutations in EF hands on Ca$^{2+}$-dependent FRET change on the indicated K$_V$7 isoforms.

**Figure supplement 2—source data 1.** Tabulated FRET values for each condition.

## Ca$^{2+}$ binding to EF3 is critical for signaling

Wild-type or mutant CaMs were co-expressed with the K$_V$7.2 CRD in bacteria, the 1:1 complex was purified, and Ca$^{2+}$ signaling was examined by monitoring the transfer of energy between the two fluorophores attached to the N- and C-termini of the AB fork with a flexible linker (see inset in *Figure 2*). This flexibility favors that FRET efficiency would be mainly proportional to packing of hA and hB. FRET efficiency was reduced in a Ca$^{2+}$ concentration-dependent manner as previously described (*Bernardo-Seisdedos et al., 2018*). Mutations into EF1 and EF2 (CaM12) did not significantly alter Ca$^{2+}$-dependent signaling (n=6), whereas mutations at either EF-hands 3 or 4 (CaM3 or CaM4) reduced the magnitude of FRET changes (n=6). The extent of the effect was significantly decreased in the complex with CaM3, with a minor effect in the complex with CaM4 (*Figure 2*).

The role of EF3 was further examined combining Ca$^{2+}$-binding canceling mutations in EF1, EF2, and EF4-hands. The AB/CaM124 complex, that is, with only EF3 able to bind Ca$^{2+}$, presented a response to Ca$^{2+}$ that was ~80% that of the AB in complex with WT CaM (n=6). In contrast, the response of the complex with CaM3 was reduced to ~30% (n=6). A similar strategy was followed to evaluate the role of the EF4 hand, testing complexes with CaM123 and CaM4. In the complex with CaM123, the response was almost abolished, whereas in the complex with CaM4, the response was about 90% of that of WT (n=6; *Figure 2*). Similar results were obtained with complexes between the CDRs of the

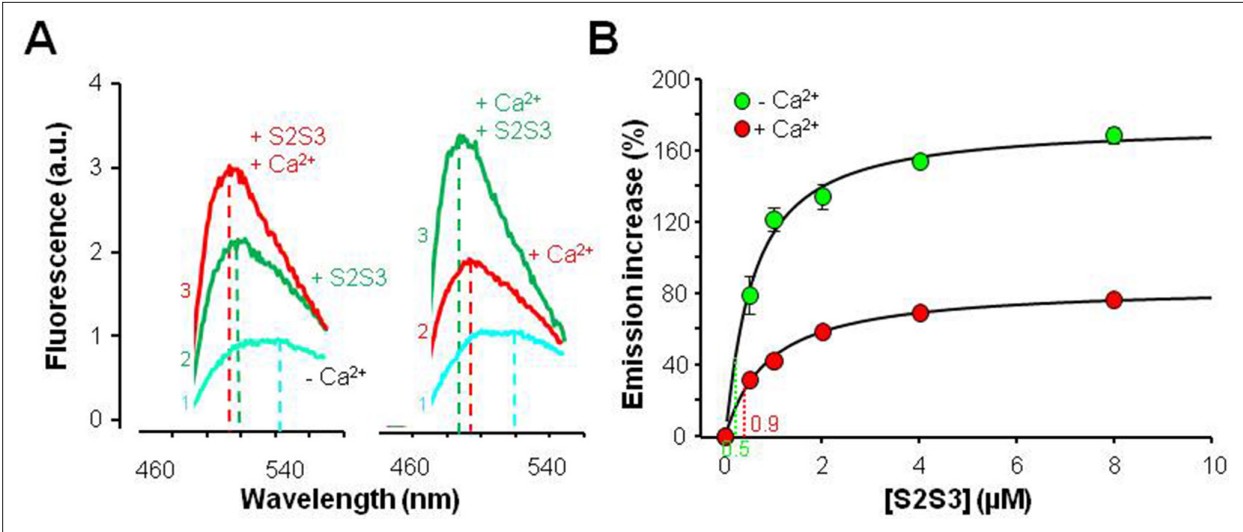

**Figure 3.** Effects of a 24 residues $K_V7$ S2S3 peptide on fluorescence emission of dansylated calmodulin (D-CAM). (**A**) Emission spectra of D-CaM (50 nM) in $Ca^{2+}$-free conditions (cyan), and after subsequent sequential addition of the S2S3 peptide (16 µM, green), and $Ca^{2+}$ (10 µM free concentration, red). The order of additions is indicated at the left of each trace. (**B**) Dose-dependent relative fluorescent emission increase as a function of S2S3 peptide concentration, in the absence (green) and the presence of $Ca^{2+}$ (10 µM, red). For this purpose, the maximum fluorescence D-CaM emission was measured between 490 and 500 nm and normalized with respect to the reference value (D-CaM with no added $Ca^{2+}$ [green] and D-CaM with 10 µM free $Ca^{2+}$ [red]). A Hill equation was fitted to the data (continuous line) with $EC_{50}=0.88 \pm 0.12$ and $1.63\pm0.07$ µM, in the absence and the presence of $Ca^{2+}$, respectively. The $K_V7.1$ S2S3 peptide sequence was Ac-RLWSAGCRSKYVGVWGRLRFARK-NH$_2$.

The online version of this article includes the following source data and figure supplement(s) for figure 3:

**Source data 1.** Spectra data for the indicated conditions, and tabulated peak values.

**Figure supplement 1.** Relative disposition of calmodulin EF3 and S2S3 loop from different $K_V7$ channel/CaM complexes solved by cryo-EM.

**Figure supplement 2.** Effect of $K_V7$ S2S3 peptides on fluorescence emission of dansylated calmodulin (D-CaM).

remaining $K_V7$ family members and mutant CaM (*Figure 2—figure supplement 2*). Thus, EF3 plays a significant role in transmitting $Ca^{2+}$ signals to the AB fork, and EF4 plays a secondary function.

## Peptides derived from the $K_V7$ S2S3 loop interact with CaM

A subset of cryo-EM $K_V7.1$ channel particles has revealed a likely interaction between the S2S3 loop of the channel voltage sensor and the EF3 of CaM (*Sun and MacKinnon, 2017*; *Sun and MacKinnon, 2020*; *Figure 3—figure supplement 1*), which, in turn, is engaged to the AB fork. Similar arrangements were reported for the $K_V7.2$ CRD (*Bernardo-Seisdedos et al., 2018*). These structural studies suggest that the privileged role of EF3 may derive from constrains imposed by the channel architecture. To address the significance of this interaction in the absence of other channel domains, changes in the fluorescent emission of dansylated CaM (D-CaM) produced by peptides derived from the $K_V7$ S2S3 sequence were monitored (*Alaimo et al., 2013*; *Supplementary file 1*). Interaction of alpha helices within the groove of the CaM lobes results in an increase in fluorescent emission of D-CaM, whereas the binding of $Ca^{2+}$ to the EF-hands causes, in addition to an increase in fluorescence, a leftward shift in the position of the peak in the emission spectrum (*Alaimo et al., 2013*).

The response to S2S3 peptides rendered an analogous profile to that of $Ca^{2+}$: a leftward shift on the emission peak and an increase in fluorescent emission (*Figure 3*). A similar response was observed when S2S3 peptides derived from the sequence of human $K_V7.1$ thought $K_V7.5$ were tested (n=3; *Figure 3—figure supplement 2*). The relative increase in emission intensity was twice as large in the absence of $Ca^{2+}$ (*Figure 3*). This is in contrast to what has been observed for peptides or targets that are embraced within the CaM lobes, in which the relative increase is similar with and without $Ca^{2+}$ (*Alaimo et al., 2014*; *Bonache et al., 2014*). Interestingly, the leftward shifts caused by $Ca^{2+}$ and the peptide were additive (*Figure 3*). These results suggest that $Ca^{2+}$ and the peptide can interact with CaM simultaneously and that $Ca^{2+}$ mitigates the effect of the peptide on D-CaM.

## NMR reveals interaction of the S2S3 peptide with the C-lobe of the AB/CaM complex

The NMR signals from labeled WT CaM complexed with non-labeled $K_V$7.2 AB fork were compared in the presence and absence of the S2S3 peptide and with $Ca^{2+}$ added (holo-CaM, four EF-hands $Ca^{2+}$-loaded) or not added (int-CaM, N-lobe $Ca^{2+}$-loaded). Chemical shift perturbations (CSPs) produced by the S2S3 peptide (13 equivalents) in the $^1H$-$^{15}N$-HSQC map of int-CaM (holo-N-lobe and apo-C-lobe) and holo-CaM in complex with the $K_V$7.2 CRD are shown in *Figure 4A* (see also *Figure 4—figure supplement 1*). In the presence of the S2S3 peptide, several resonances of CaM residues in the spectrum were shifted, most of them located in the C-lobe. The CSPs perturbations, color-coded in the structure of the human $K_V$7.2 CRD in *Figure 4—figure supplement 1*, are consistent with the S2S3 loop interacting predominantly with the EF3 loop, both in absence and in the presence of $Ca^{2+}$. EF3 displacements were observed for D94, N98, Y100, I102, and A104, whereas for EF4, changes in the environment of I131 and E139 are beyond the threshold level (*Figure 4B*). Thus, $Ca^{2+}$ addition produces a significant perturbation map, which is in line with the differential relative increase in fluorescence caused by the peptide in the D-CaM assay (*Figure 3B*). Next, we performed atomistic molecular dynamics (MDs) simulations to investigate the interactions between the $K_V$7.1 S2S3 peptide and int- or holo-CaM in complex with the $K_V$7.2 CRD (*Supplementary file 2*). Consistent with the NMR interaction experiment, the contact map obtained from the simulations shows that the peptide interacts mainly with the EF3 loop and the linker connecting CaM lobes (*Figure 4C–D* and *Figure 4—figure supplement 2*). In contrast, there were no contacts in the region connecting EF3 and EF4, suggesting that the CSPs observed are better interpreted as an allosteric effect, rather than a direct contact with the peptide.

Regarding the S2S3 peptide, residues that form an intracellular loop located between W166 and G176 are the ones that interact predominantly with CaM (*Figure 4D*). During the course of the simulation, the C-terminal region adopted an α-helix conformation for ≥97.8% of the time (see *Figure 4—figure supplement 3*). The N-terminal that started as a $3_{10}$ helix became unstructured after the initial equilibration. It is reasonable to expect such differential stability since the N- and C-helices were initially formed by 6 and 10 residues, respectively, and $3_{10}$ helices are less stable than α-helices (*Bolin et al., 1999*).

The interaction between S2S3 and EF3 was more stable when it was not loaded with $Ca^{2+}$ (*Figure 4F*, left). In contrast, the main contacts of the holo-system were established primarily with the linker connecting CaM lobes (*Figure 4F*, right). To analyze the interaction between S2S3 and EF3, we measured the distance between the center of mass of EF3 and S2S3 loops or the linker connecting the CaM lobes. The results suggest that the interaction between the S2S3 and the empty EF3 loops is rather stable, whereas $Ca^{2+}$ occupancy prompts the movement of the peptide away from EF3 toward the linker on the lobes (*Figure 4E*). Thus, $Ca^{2+}$ occupancy has an important influence on the S2S3/CaM interaction. These observations could fully explain the reduced CaM/S2S3 affinity in the presence of $Ca^{2+}$ (see *Figure 3*).

## Reversal of $Ca^{2+}$-EF3 signaling by S2S3 peptides

Changes on FRET index in response to $Ca^{2+}$ in the presence of S2S3 peptides were monitored as previously described (see *Figure 2*). *Figure 5* shows that the $Ca^{2+}$-dependent reduction in FRET index was mitigated as the concentration of peptide was increased. At high-peptide concentrations (≥10 μM), the FRET index increased, suggesting that the distance/orientation of AB helices was even more favorable than in the absence of $Ca^{2+}$. A similar behavior was observed when the effect of the peptide for the other $K_V$7 family members was examined (*Figure 5—figure supplement 1*).

S2S3 is not canceling the effect of $Ca^{2+}$ by competing or displacing this cation from its binding site. Instead, the response to $Ca^{2+}$ in the presence of S2S3, in terms of FRET index, was in the opposite direction than when the peptide was absent. The magnitude of signaling reversal was similar in WT and CaM124 complexes, whereas it was reduced in complexes with CaM3 (*Figure 5*). Thus, the direction/orientation of the movements in the AB fork when EF3 is loaded with $Ca^{2+}$ is reversed upon interaction with S2S3.

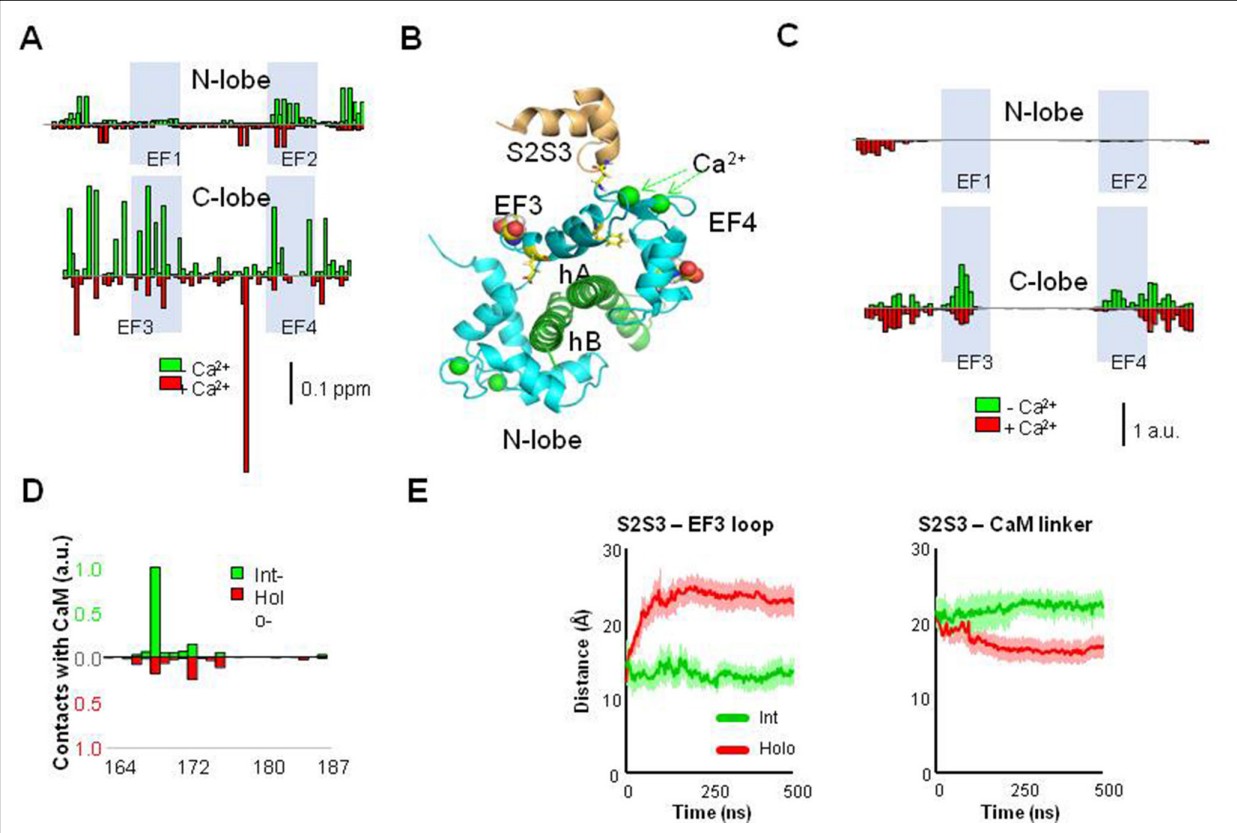

**Figure 4.** Interaction between the K$_v$7 S2S3 peptide and CaM in complex with K$_v$7.2 CDR. (**A**) The chemical shift perturbation (CSP) analysis shows that the magnitude of local residue environmental alterations detected by NMR is larger in the C-lobe, both in the presence and in the absence of Ca$^{2+}$. (**B**) Structural mapping of the main CSPs in the presence of Ca$^{2+}$ over Ca$^{2+}$-loaded K$_v$7.2 CaM/CDR complex. The two resides with the larger displacements are represented as balls, whereas the remaining above three times the mean are represented as sticks. The structure of the S2S3 loop was derived from the Cryo-EM PDB 5VMS (**Sun and MacKinnon, 2017**) and placed according to structural alignment of the C-lobe of PDB 6FEH (**Bernardo-Seisdedos et al., 2018**). (**C**) Contact map derived from molecular dynamic (MD) simulations of the S2S3/CaM complex. Normalized CaM contacts with the S2S3 peptide residues (10 Å cut-off) for int- (green) and holo-systems (red; see **Figure 5—figure supplement 1**). Vertical calibration bar is in arbitrary units (a.u.). (**D**) S2S3 contact map with CaM residues (4 Å cut-off; see **Figure 5—figure supplement 1**). (**E**) Distance as a function of time between the mass centers of the EF3 loop (residues D93-G98) and (i) the S2S3 loop (residues R164-L173; left) or (ii) the linker connecting CaM lobes (residues R74-E84; right). Bars indicate SEM (n=6).

The online version of this article includes the following source data and figure supplement(s) for figure 4:

**Source data 1.** Tabulated data values for NMR chemical shift perturbations.

**Source data 2.** NMR raw spectra of KV7.2/Calmodulin complex with and without calcium in presence of S2S3 peptide.

**Source data 3.** Replica 1: Molecular dynamic trajectory of holo system, calcified calmodulin in presence of S2S3.

**Source data 4.** Replica 2: Molecular dynamic trajectory of holo system, calcified calmodulin in presence of S2S3.

**Source data 5.** Replica 3: Molecular dynamic trajectory of holo system, calcified calmodulin in presence of S2S3.

**Source data 6.** Replica 4: Molecular dynamic trajectory of holo system, calcified calmodulin in presence of S2S3.

**Source data 7.** Replica 5: Molecular dynamic trajectory of holo system, calcified calmodulin in presence of S2S3.

**Source data 8.** Replica 6: Molecular dynamic trajectory of holo system, calcified calmodulin in presence of S2S3.

**Source data 9.** Replica 6: Molecular dynamic trajectory of holo system, calcified calmodulin in presence of S2S3.

**Source data 10.** Replica 1: Molecular dynamic trajectory of int system, calcified N-lobe (no calcium C-lobe) of calmodulin in presence of S2S3.

**Source data 11.** Replica 2: Molecular dynamic trajectory of int system, calcified N-lobe (no calcium C-lobe) of calmodulin in presence of S2S3.

**Source data 12.** Replica 3: Molecular dynamic trajectory of int system, calcified N-lobe (no calcium C-lobe) of calmodulin in presence of S2S3.

**Source data 13.** Replica 4: Molecular dynamic trajectory of int system, calcified N-lobe (no calcium C-lobe) of calmodulin in presence of S2S3.

**Source data 14.** Replica 5: Molecular dynamic trajectory of int system, calcified N-lobe (no calcium C-lobe) of calmodulin in presence of S2S3.

**Source data 15.** Molecular dynamic trajectory of S2S3 peptide y solution.

*Figure 4 continued on next page*

*Figure 4 continued*

**Figure supplement 1.** $^{15}$N-HSQC of $K_V$7.2 CDR:CaM titrated with unlabeled $K_V$7.1-S2S3 in absence (left) and in presence of 1 mM $Ca^{2+}$.

**Figure supplement 1—source data 1.** HSQC of KV7.2 CDR:CaM titrated with unlabeled KV7.1-S2S3 in absence of 1 mM Ca2+.

**Figure supplement 2.** Contact maps derived from molecular dynamic (MD) simulations of the S2S3/CaM complex.

**Figure supplement 2—source data 1.** HSQC of KV7.2 CDR:CaM titrated with unlabeled KV7.1-S2S3 in presence of 1 mM Ca2+.

**Figure supplement 3.** CD spectra of S2S3 peptide before and after treatment with H2O2.

**Figure supplement 3—source data 1.** Raw data for CD spectra of S2S3 peptide before and after treatment with H2O2, and tabulated values of percentage helicity vs position.

## Treatment with $H_2O_2$ reduces the effect of the S2S3 peptide

The S2S3 loop, which is highly conserved among $K_V$7 channels, contains one ($K_V$7.1) or three cysteine residues ($K_V$7.2–$K_V$7.5, *Figure 3—figure supplement 1*). The cysteine site mediates an increase in channel open probability in response to oxidizing conditions (*Gamper et al., 2006*). We tested the influence of oxidation by removing DTT (Dithiothreitol) from the buffer and including $H_2O_2$ to obtain a derivate that will be referred to as oxidized-S2S3. Contrary to the increase observed with S2S3, no changes in fluorescent emission of D-CaM were observed after addition of oxidized-S2S3 (*Figure 6—figure supplement 1*).

Treatment with $H_2O_2$ or DTT did not affect the response of the $K_V$7.4AB/CaM or $K_V$7.2AB/CaM complexes to $Ca^{2+}$ (*Figure 6—figure supplement 2*). The $Ca^{2+}$ titration profile using $K_V$7.4AB/CaM or $K_V$7.2AB/CaM complexes in the presence of oxidized $K_V$7.4 or $K_V$7.2 peptides (10 µM) was similar to that obtained in the absence of S2S3, suggesting that oxidized S2S3 can no longer affect the AB-CaM interaction (*Figure 6*). Using the $K_V$7.4 sensor and oxidized $K_V$7.4-S2S3 peptide, the relative

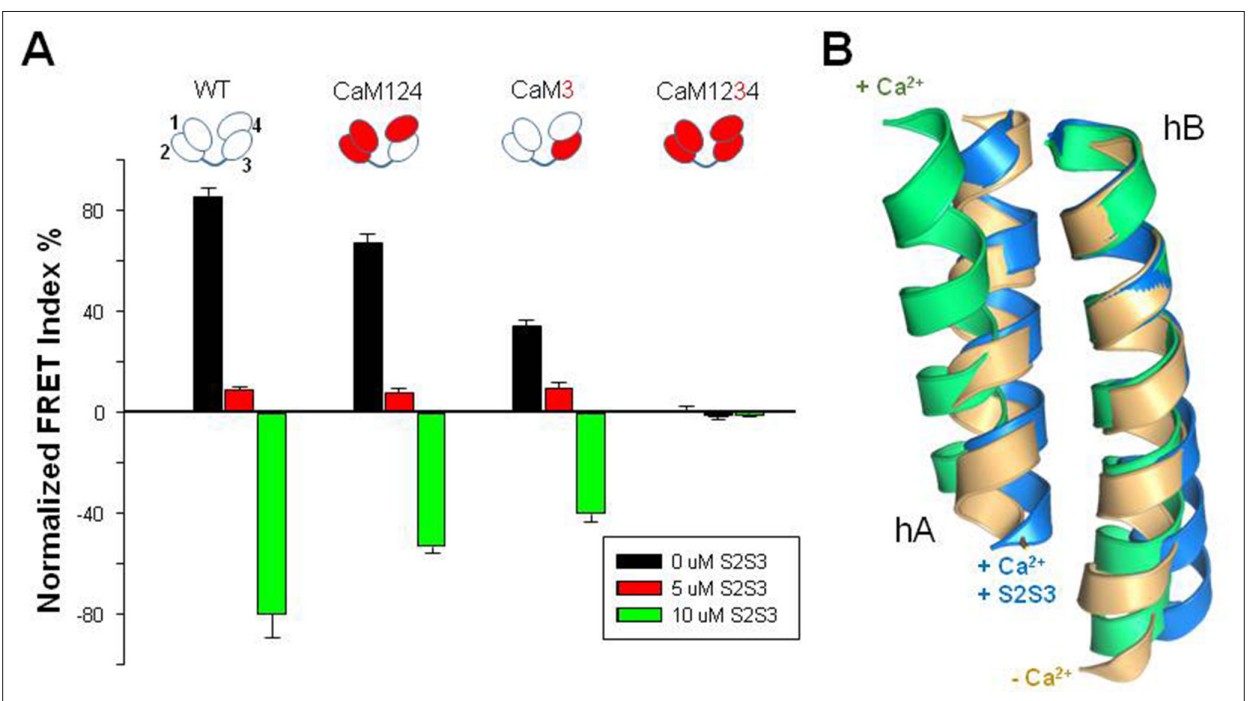

**Figure 5.** Relative FRET changes of the human $K_V$7.2 in complex with mutant CaM. (**A**) FRET change after $Ca^{2+}$ addition (10 µM) in the presence of the indicated concentrations of the S2S3 peptide: control (black), 0 µM S2S3, 5 µM S2S3 (red), and 10 µM S2S3 (green) (n=4). (**B**) Superposition of helices A and B solved in the absence of $Ca^{2+}$ (gold, PDB 6FEH, $K_V$7.2), in the presence of $Ca^{2+}$ (green, PDB 6FEG, $K_V$7.2), and interacting with the S2S3 loop in the presence of $Ca^{2+}$ (blue, PDB 5VMS, $K_V$7.1).

The online version of this article includes the following source data and figure supplement(s) for figure 5:

**Source data 1.** Tabulated FRET values for each condition.

**Figure supplement 1.** Relative FRET changes of $K_V$7 CRD in complex with CaM.

**Figure supplement 1—source data 1.** Tabulated FRET values for each condition.

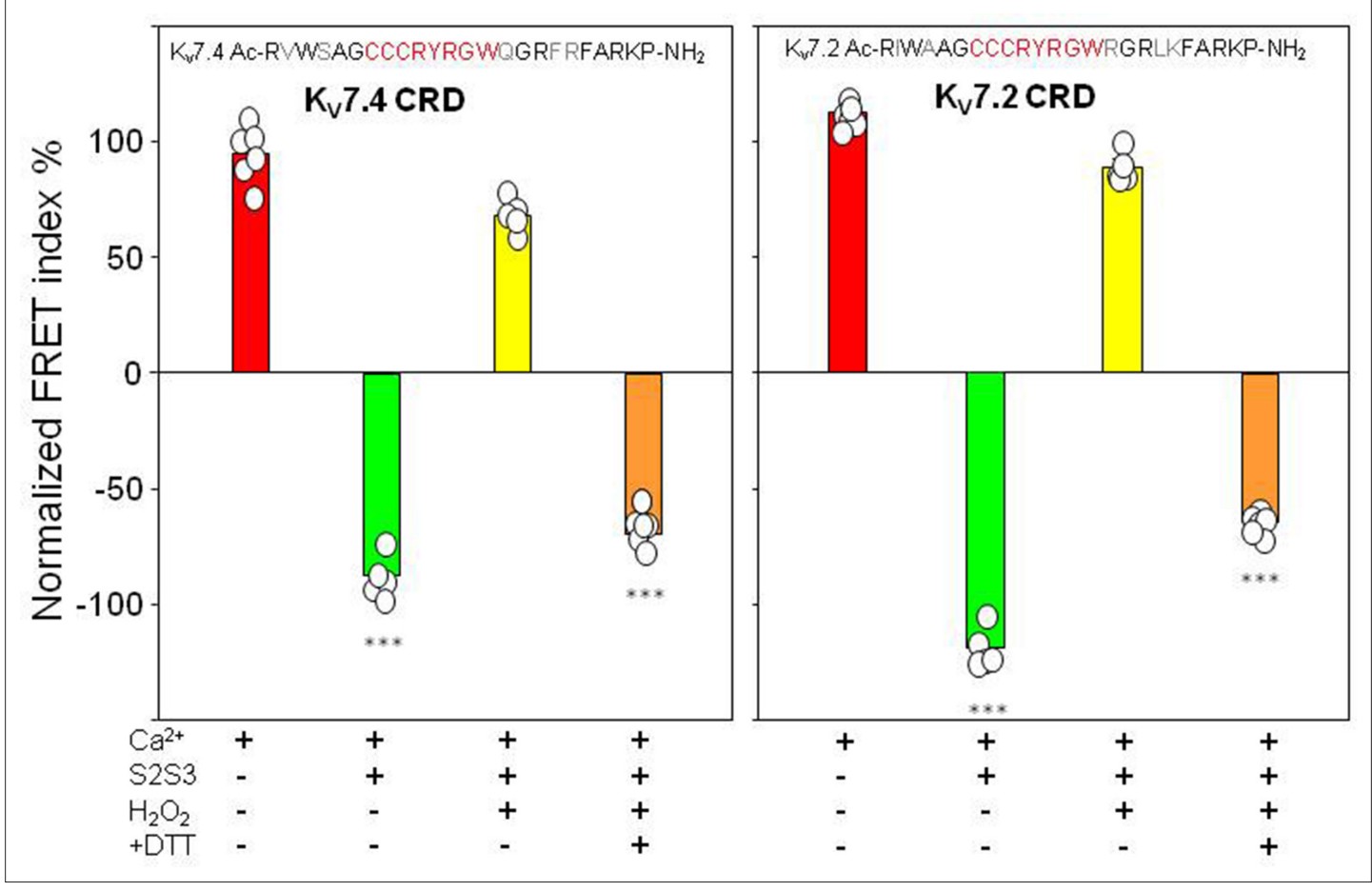

**Figure 6.** FRET efficiency changes prompted by oxidized and reduced S2S3 peptides. Difference in FRET efficiency in the absence and the presence of $Ca^{2+}$. Left, $K_V7.4$-S2S3 peptide and $K_V7.4$ CRD. Right, $K_V7.2$-S2S3 peptide and $K_V7.2$ CRD. Similar results were obtained with $K_V7.1$-S2S3 peptide (**Figure 6—figure supplement 2**). Red, sensor alone. Green, in the presence of 10 µM peptide. Yellow, with 10 µM oxidized peptide. Orange, oxidized peptide treated with 1 mM DTT. Bars represent mean ± SEM FRET-efficiency. *p<0.05; ***p<0.001. Each plot represents the average of at least six independent experiments.

The online version of this article includes the following source data and figure supplement(s) for figure 6:

**Source data 1.** Tabulated FRET values for each condition.

**Figure supplement 1.** Fluorescent emission D-CaM enhancement caused by $K_V7.1$ S2S3 and $K_V7.2$ S2S3 peptides before and after oxidation.

**Figure supplement 1—source data 1.** Tabulated fluoresce emission values for each condition.

**Figure supplement 2.** Percentage of FRET index changes prompted by oxidized and reduced S2S3 peptides.

**Figure supplement 2—source data 1.** Tabulated FRET values for each condition.

FRET change in presence of $Ca^{2+}$ was 68.22±6.90, i.e., a change qualitatively similar to that obtained in the presence of $Ca^{2+}$ alone (94.84±5.93), and very different from that obtained with the untreated peptide (–87.50±8.26). The oxidized S2S3 peptide was incubated with the reducing agent DTT aiming to reverse the effect of the treatment with $H_2O_2$. We observed a partial recovery: after treating the oxidized peptide with DTT, the relative FRET change was similar to that observed with the untreated peptide (–69.45±12.80; **Figure 6**, left panel). Similar results were obtained using the $K_V7.2$ sensor and the $K_V7.2$-peptide (**Figure 6**, right panel) or when using the $K_V7.1$-peptide (**Figure 6—figure supplements 1 and 2**).

## All $K_V7$ CRDs display a similar response to $Ca^{2+}$

As mentioned above, the effect of S2S3 peptides having the sequence of each $K_V7$ isoform was tested in D-CaM assay, obtaining equivalent results for the set. The apparent affinity for every peptide was significantly lower in the presence of $Ca^{2+}$ (**Figure 3—figure supplement 2**). A panel of $K_V7$ biosensors

in which the fork sequence was replaced by the equivalent segment from $K_V7.1$, $K_V7.2$, $K_V7.3$, $K_V7.4$, and $K_V7.5$ human isoforms was created. FRET index was reduced for all biosensors in the presence of $Ca^{2+}$, and the signal was significantly preserved in complexes formed with CaM124, whereas it was decreased in complexes with CaM3. The apparent affinity for $Ca^{2+}$ was lower in the presence of the peptides, but the difference was not statistically significant (*Figure 5—figure supplement 1*). Thus, we conclude that EF3 plays a similar role across the $K_V7$ family of CRDs.

## Discussion

As its name suggests, CaM is a CALcium MODULated protein, regarded as a fundamental player in the orchestration of $Ca^{2+}$ signals in every eukaryotic cell (*Clapham, 2007*). Notwithstanding, its function is not limited to $Ca^{2+}$ signaling, having important functions in protein trafficking to the plasma membrane, protein folding, and other functions (*Villarroel et al., 2014*). Here, the portfolio of CaM capacities is extended by providing evidence of its essential role in transducing redox signaling in conjunction with $K_V7$ channels, which exhibit an exquisite sensitivity to oxidation. Current augmentation can be induced by an external $H_2O_2$ concentration as low as 5 μM or nM concentrations when using a faradaic device for delivery (*Abdullaeva et al., 2022*; *Gamper et al., 2006*), which is within the range of extracellular peroxide release due to dopamine oxidation in rat brain (*Kulagina and Michael, 2003*). Such high sensitivity places $K_V7$ channels among candidate proteins that are first to respond to $H_2O_2$ production and reveals a prominent role of CaM in redox signaling.

Previously, we demonstrated the significance of cysteine residues in neuronal $K_V7$ channels located in the unusually long (compared to other voltage-operated $K^+$ channels) intracellular linker joining transmembrane segments S2 and S3, which are part of the voltage sensor (*Gamper et al., 2006*). Here, we show that the ability of the EF3-hand of CaM to bind $Ca^{2+}$ is essential in this redox-signaling pathway. This is inferred, among others, from the observation that the effect of bath application of $H_2O_2$ is lost in cells overexpressing CaM variants with a disabled EF3-hand or treated with the membrane permeable $Ca^{2+}$ chelator BAPTA-AM.

Earlier, we showed that the redox response depends on the presence of cysteine residues at the S2S3 loop (*Gamper et al., 2006*); yet, the number of cysteine residues is not the sole factor defining the efficacy of the response. No evidence for redox regulation was observed for WT $K_V7.1$ channels (that have only one cysteine residue at the S2S3 loop); and engineered 'three Cys $K_V7.1$ channel' displayed a weak response to $H_2O_2$. WT $K_V7.4$ (three Cys residues in the S2S3) displayed a strong response to $H_2O_2$, yet a partial response was still observed for engineered 'one Cys $K_V7.4$ channels' (*Gamper et al., 2006*). Hence, even with only one cysteine present, the S2S3 linker can mediate $H_2O_2$ sensitivity of a 'responsive' $K_V7$ channel (such as $K_V7.4$), while there must be other structural constrains that hinder potentiation of $K_V7.1$ by the $H_2O_2$.

We suggest that a low channel open probability ($p_o$) is an additional requirement for redox sensitivity because no evidence for redox regulation has been observed for $K_V7.3$ channels, which present a $p_o$ close to the unity at saturating voltages (*Li et al., 2004*). These channels have a triple Cys residue motive at the S2S3 loop that only differs at one position when compared to redox-sensitive $K_V7.2$ channels (an Arg residue in $K_V7.3$ versus a Lys residue in $K_V7.2$). We suggest that due to very little room for further increase in $p_o$, oxidation has negligible consequences on macroscopic $K_V7.3$ currents.

The redox response is characterized by a remarkable increase/recovery in M-current density on a second to minute time scale, and it can be reversed by reducing agents (*Abdullaeva et al., 2022*; *Gamper et al., 2006*). This could derive from insertion of new channels, engaging silent channels, higher open probability, or a combination of these. Although the insertion/recruitment of new channels cannot be completely discarded, $H_2O_2$ causes an increase in single-channel activity in excised patches where the incorporation of new channels cannot take place. The redox impact on $K_V7$ channels is accompanied by a left shift in the current-voltage relationship of macroscopic currents, meaning that channel opening becomes easier at lower voltages or that the probability of opening at a given voltage increases (*Gamper et al., 2006*). Large leftward shifts in voltage dependency of $K_V7$ channels after over-expression of CaM C-lobe mutants (*Chang et al., 2018*; *Sihn et al., 2016*) have been reported, suggesting a critical role for EF3 (*Chang et al., 2018*). However, this shift has not always been observed (*Archer et al., 2019*). In this study, we saw relatively small (~10 mV) but consistent leftward shift in voltage dependence of $K_V7.4$ co-expressed with all CaM mutants containing the EF3 mutation, suggesting a degree of tonic channel inhibition conferred via EF3.

Taking into account the images at atomic resolution of CaM interacting with the voltage sensor of $K_V7.1$ channels (*Sun and MacKinnon, 2017*), it is tempting to propose that through this interaction, CaM is dragging the voltage sensor, making it more difficult to reach the up position that leads to gate opening in response to depolarizations, or CaM is stabilizing the state where the voltage sensor is disengaged from the pore. In other words, we envisage CaM constitutively and dynamically inhibiting channel activation and that the redox action relieves this inhibition by weakening the dynamic CaM-S2S3 interaction in a $Ca^{2+}$-dependent manner. This idea fits with the observation that EF3 is not interacting with S2S3 in the available atomic resolution structures trapped in a partially (*Sun and MacKinnon, 2020*) and fully (*Zheng et al., 2021*) open states. To harmonize with other observations, we propose that this inhibition is counterbalanced by CaM-dependent promotion of surface expression when CaM or CaM1234 are over-expressed (*Etxeberria et al., 2008*; *Gomis-Perez et al., 2017*). Furthermore, our data suggest that $Ca^{2+}$ binding to EF3 should help releasing the voltage sensor from CaM. In principle, this should lead to current potentiation, but the effect of $Ca^{2+}$ on gating may be conditioned by other circumstances, as discussed next.

To what extend the observations can be extrapolated to the native channel is a concern raised when using peptides. Our MD simulations reveal that the N-terminal helix present in the channel structure of the S2S3 domain disappears when the peptide is in solution. In contrast, the C-terminal helix is remarkably stable. Nevertheless, the map of interactions between EF3 and CaM highlights the loop connecting these two helices, which overlap with the residues found or assumed to make contacts in the $K_V7$/CaM complexes (*Sun and MacKinnon, 2017*). However, conclusions derived from the use of isolated peptides may not translate linearly to the whole channel complex. Interestingly, FRET changes in the isolated recombinant CRD caused by S2S3 peptides depend on the concentration of $Ca^{2+}$ are reversibly sensitive to $H_2O_2$ treatment and primarily governed by EF3. The left-shift observed on the emission spectrum of D-CaM, a property observed only after loading EF-hands with $Ca^{2+}$ (*Alaimo et al., 2013*), is consistent with a functional interaction of the peptides with the EF-hands in the loops coordinating $Ca^{2+}$. The dose-response relationship with D-CaM and our MD simulations illustrates that the interaction is weaker when the EF3-hand is loaded with this cation. It is very clear that the influence of S2S3 on the relative orientation of helices A and B is only manifested in the presence of $Ca^{2+}$, which is also a necessary condition for functional effects of $H_2O_2$ on $K_V7.4$ currents. This is remarkable because S2S3 interacts with apo-CaM causing a shift on the peak emission of D-CaM (i.e. presumably, through EF-hands). The lack of any impact on energy transfer on the CRD reveals that the interaction under low $Ca^{2+}$ conditions does not result in a conformational change in the AB fork. The additive increase in the magnitude of D-CaM fluorescent emission and left-shift in peak emission is indicative of the concurrent interaction of $Ca^{2+}$ and the S2S3 loop with EF3, something that becomes apparent when examining the $K_V7.1$/CaM complex (*Sun and MacKinnon, 2017*).

Remarkably, we find that EF3 is essential within the isolated recombinant CRD to translate $Ca^{2+}$ signaling into conformational changes, which result in an 18° opening of the AB fork (*Bernardo-Seisdedos et al., 2018*). Thus, signaling through EF3 is a property inherent to the CRD and does not derive from constrains that the geometry of the voltage sensor-pore imposes. We note that the architecture of $K_V7$ channels allows exploiting EF3 signaling in a more efficient way. We call attention on the conditional duality of this signaling system. One branch operates on the voltage sensor (S2S3/EF3), and another branch changes the orientation of helix A, likely affecting S6, and therefore, the main gate formed by the S6 bundle crossing. Whereas we propose that the S2S3/EF3 branch inhibits the current by disengaging the voltage sensor, the consequences of the relative movements of helix A are unclear. Helix A and S6 are connected by a conserved flexible linker whose helical character varies upon $PIP_2$ binding, causing a variable S6-hA orientation relative to the gate (*Li et al., 2021a*; *Li et al., 2021b*; *Niu et al., 2020*; *Sun and MacKinnon, 2020*; *Zheng et al., 2021*). The conditional S2S3-dependent reorientation of helix A caused by $Ca^{2+}$ is posited to favor or hinder pore opening depending on the initial orientation of the CRD, which, in turn, depends on $PIP_2$ occupancy (*Li et al., 2021b*; *Niu et al., 2020*; *Zheng et al., 2021*). This complexity of interactions could explain contrasting effects of $Ca^{2+}$ elevation. Whereas in rat superior cervical neurons (*Selyanko and Brown, 1996*) and CHO cells (*Gamper and Shapiro, 2003*; *Kosenko and Hoshi, 2013*), this cation inhibits M-currents, a clear current potentiation was observed in *Xenopus* oocytes (*Gómez-Posada et al., 2011*). Further experiments are required to clarify this issue. We note that our data are compatible with the proposed role of S2S3 and EF3 loops in $Ca^{2+}$ $K_V7.4$ regulation (*Zhuang and Yan, 2020*) and highlight an intricate

conditional network of signaling processes, in which $PIP_2$ binding, redox regulation, and $Ca^{2+}$ are interconnected (**Gomis-Perez et al., 2017**).

An unexpected observation was the reversal in FRET index observed at higher S2S3 peptide concentrations only in the presence of $Ca^{2+}$. Although our data suggest that S2S3 interacts with CaM in the presence and absence of $Ca^{2+}$, occupation EF3 by this cation is a requirement to signal the reorganization of the CRD and for the functional redox effect on the channel. FRET changes caused by $Ca^{2+}$ are opposed in the presence of S2S3 peptides. At the molecular level, we do not know what the reversal of the signal implies because any modification in distance or orientation causes FRET alterations. It seems reasonable to propose that, in this scenario, the movement within the AB helices goes in the opposite direction, leading to a tighter packing of the fork, which is consistent with increased FRET signal. Tighter packing is what is observed when comparing the $K_V7.2$ fork in absence of S2S3 - with and without $Ca^{2+}$- with the S2S3 loop interacting with EF3, presumably loaded with $Ca^{2+}$ (**Sun and MacKinnon, 2017**; **Figure 5B**).

Given the structural similarities between the CaM lobes, it is remarkable that both the CSPs maps and the MD contact maps report mainly signals from the C-lobe. In reference to the MD data, it should be borne in mind that the initial condition corresponds to S2S3 interacting with EF3, which introduces a bias in favor of capturing interactions with the C-lobe. Regarding the NMR data, two observations provide clues that help explaining this apparent selectivity. On one hand, the displacements are, in general, of smaller magnitude in the presence of $Ca^{2+}$. Incidentally, this goes along with the observation that the relative fluorescence signal from D-CaM with S2S3 and $Ca^{2+}$ is about half of what is observed when $Ca^{2+}$ is chelated. On the other hand, under basal conditions, the N-lobe sites are already occupied by this cation (**Bernardo-Seisdedos et al., 2018**), and therefore, the displacements are expected to be smaller to begin with. Nevertheless, even under high $Ca^{2+}$ conditions (i.e. with the four EF-hands occupied), the overall displacements are observed preferentially at the C-lobe, consistent with an intrinsic preference for this lobe. Regarding the displacements observed at the N-lobe, a similar magnitude under the two experimental conditions explored could be expected since $Ca^{2+}$ occupies the two sites at the N-lobe in both situations. However, there are some differences, especially at the initial portion of the N-terminus and at the EF2 binding loop. In addition, there is a notable difference between the MD contact map and the NMR CSPs within the residues connecting EF3 and EF4. We speculate that these differences are a consequence of allosteric EF-hand-coupling (**Sorensen and Shea, 1998**).

Based on docking calculations, the functional existence of significant interactions between a target protein and apo-CaM through the EF-hands of the C-lobe was first proposed for the smoothelin-like 1 protein (**Ishida et al., 2008**). Subsequently, direct interactions between apo-EF3 and myosin were observed in X-ray structures, and this interaction was postulated to play an important role in signal transduction (**Münnich et al., 2014**). Recent analysis of surface interaction from cryo-EM structures of ion channels with CaM hints to interactions between EF3 and EF4 with Eag1, TRPV5, and TRPV6 channels (**Núñez et al., 2020**). Thus, $Ca^{2+}$ signal bi-directional transduction through direct interaction between the $Ca^{2+}$-binding sites and CaM target protein could be a widespread mechanism.

## Materials and methods

**Key resources table**

| Reagent type (species) or resource | Designation | Source or reference | Identifiers | Additional information |
|---|---|---|---|---|
| Peptide, recombinant protein | $K_V7.1$ S2S3 | Proteogenix | | RLWSAGCRSKYVGVWG RLRFARKP |
| Peptide, recombinant protein | $K_V7.2$ S2S3 | Proteogenix | | RIWAAGCCCRYRGWRG RLKFARKP |
| Peptide, recombinant protein | $K_V7.3$ S2S3 | Proteogenix | | RIWAAGCCCRYRKGWR LFKFARKP |
| Peptide, recombinant protein | $K_V7.4$ S2S3 | Proteogenix | | RVWSAGCCCRYRGWQG RFRFARKP |

*Continued on next page*

*Continued*

| Reagent type (species) or resource | Designation | Source or reference | Identifiers | Additional information |
|---|---|---|---|---|
| Peptide, recombinant protein | $K_V$7.5 S2S3 | Proteogenix | | RIWSAGCCCRYRGWQG RLRFARKP |
| Recombinant DNA reagent | $K_V$7.1 mtfp-hAB-Venus (residues I247-D456) in pPROEX HTc | This paper | NM_000218.2 | Plasmid, Fluorescence sensor |
| Recombinant DNA reagent | $K_V$7.2 mtfp-hAB-Venus (residues I310-D549), in pPROEX HTc vector | This paper | NM_172107.3 | Plasmid, Fluorescence sensor |
| Recombinant DNA reagent | $K_V$7.3 mtfp-hAB-Venus (residues I349-D556) in pPROEX HTc | This paper | NM_004519.3 | Plasmid, Fluorescence sensor |
| Recombinant DNA reagent | $K_V$7.4 mtfp-hAB-Venus (residues I315-D539) in pPROEX HTc | This paper | NC_060925.1 | Plasmid, Fluorescence sensor |
| Recombinant DNA reagent | $K_V$7.5 mtfp-hAB-Venus (residues I308-D527) in pPROEX HTc | This paper | NC_060930.1 | Plasmid, Fluorescence sensor |
| Recombinant DNA reagent | CaM in pOKD4 | Recombinant Human CALM2 in pOKD4 vector, GenScript | Genbank, NP_001292553.1 | Plasmid, Calmodulin, human CALM2 |
| Recombinant DNA reagent | hKCNQ4-eYFPc | *Gamper et al., 2006* | AF105202 | Plasmid,KCNQ4 bound to YFP |
| Recombinant DNA reagent | hKCNQ4CCC/AAA -eYFPc | Mutant AF105202; *Gamper et al., 2006* | | Plasmid, Mutant KCNQ4 bound to YFP |
| Recombinant DNA reagent | CaM 3 in pOKD4 | GenScript | Genbank, NP_001292553.2 | Plasmid, human CALM2, D93A mutation |
| Recombinant DNA reagent | CaM124 in pOKD4 | GenScript | Genbank, NP_001292553.3 | Plasmid, mutant human CALM2, D20A/D56A/D129A |
| Recombinant DNA reagent | CaM123 in pOKD4 | GenScript | Genbank, NP_001292553.4 | Plasmid, mutant human CALM2, D20A/D56A/D93A |
| Recombinant DNA reagent | CaM12 in pOKD4 | GenScript | Genbank, NP_001292553.5 | Plasmid, mutant human CALM2, D20A/D56A |
| Recombinant DNA reagent | CaM34 in pOKD4 | GenScript | Genbank, NP_001292553.6 | Plasmid, mutant human CALM2, D93A/D129A |
| Recombinant DNA reagent | CaM1234 in pOKD4 | GenScript | Genbank, NP_001292553.7 | Plasmid, mutant human CALM2, D20A/D56A/D93A/D129A |
| Recombinant DNA reagent | CaM WT in pCDNA3 | GenScript | Genbank, NP_001292553.8 | Plasmid, Calmodulin, human CALM2 |
| Recombinant DNA reagent | CaM3 in pCDNA3 | GenScript | Genbank, NP_001292553.9 | Plasmid, human CALM2, D93A mutations |
| Recombinant DNA reagent | CaM34 in pCDNA3 | GenScript | Genbank, NP_001292553.10 | Plasmid, human CALM2, D93A/D129A mutations |
| Recombinant DNA reagent | CaM12 in pCDNA3 | GenScript | Genbank, NP_001292553.11 | Plasmid, human CALM2, D20A/D56A mutations |
| Recombinant DNA reagent | CaM124 in pCDNA3 | GenScript | Genbank, NP_001292553.12 | Plasmid, human CALM2, D20A/D56A/D129A mutations |
| Recombinant DNA reagent | CaM1234 in pCDNA3.1 | GenScript | Genbank, NP_001292553.13 | Plasmid, human CALM2, D20A/D56A/D93A/D129A mutations |

*Continued on next page*

*Continued*

| Reagent type (species) or resource | Designation | Source or reference | Identifiers | Additional information |
|---|---|---|---|---|
| Recombinant DNA reagent | $K_V7.4$ in pCDNA3.1 | GenScript | Genbank, NP_001292553.14 | Plasmid, human $K_V7.4$ channel |
| Recombinant DNA reagent | $K_V7.4$CCC/AAA in pCDNA3.1 | *Gamper et al., 2006* | Genbank, NP_001292553.15 | Plasmid, human $K_V7.4$ channel, C156A, C157A, C158V |
| Cell line (*Homo sapiens*) | Kidney (normal epithelial, embryo) | ATCC | HEK293 | |
| Chemical compound and drug | 5-(Dimethylamino) naphthalene-1-sulfonyl chloride, DNSCl | SIGMA-ALDRICH | CAS Number: 605-65-2 | Dansyl chloride |
| Chemical compound and drug | Pierce DTT (ditiotreitol) | Thermo Scientific | CAT# 20290 | DTT |
| Chemical compound and drug | Hydrogen peroxide solution | SIGMA-ALDRICH | CAS Number = 7722-84-1 / Pubchem ID = 57654227 | $H2O2$ 30% (w/w) in H2O, contains stabilizer |
| Chemical compound and drug | X2254 | SIGMA-ALDRICH | | XE-991 |
| Chemical compound and drug | ab145545 | Abcam | | Retigabine |
| Chemical compound and drug | E2311 | Promega | | Fugene |
| Software and algorithm | PyMOL | The PyMOL Molecular Graphics System, Version 1.3 Schrödinger, LLC. | | Use for molecular dynamics and figure preparation |
| Software and algorithm | Patchmaster V2 | HEKA Instruments | | |
| Software and algorithm | VMD | *Humphrey et al., 1996* | | Use for molecular dynamics and figure preparation |
| Software and algorithm | NAMD | *Phillips et al., 2020* | | Use for molecular dynamics |

## Cell culture and transfection

Human embryonic kidney line 293 (HEK293T) cells from ATCC authenticated via STR profiling by Eurofins Genomics (ISO 17025) and confirmed negative for mycoplasma via PCR (EZ-PCR Mycoplasma detection kit, Generon, 20-700-20) were cultured to 80% confluence before passaging and used for experimentation between P10 and P40. Cells were grown in Dulbecco's modified Eagle's medium (DMEM Gibco) containing penicillin (100 U/mL; Sigma), streptomycin (100 µg/mL; Sigma), and 10% fetal calf serum (Sigma). HEK293 cells were cultured in 24 well plates for 24 hr prior to transfection and transfected with 300 ng of each plasmid using FuGene transfection reagent (Promega), according to the manufacturer's instructions. Plasmids: hKCNQ4-eYFPc (AF105202) or hKCNQ4CCC/AAA -eYFPc a triple cysteine to alanine mutation in the S2–S3 linker (C156A, C157A, and C158A; *Gamper et al., 2006*) were co-transfected with WT CaM or lobe mutants. For BAPTA-AM experiments, cells were incubated with 10 µM cell-permeable BAPTA-AM (B1205 ThermoFisher) for 30 min prior to recordings. Cells were also bathed in 10 µM BAPTA-AM for the duration of the recording.

## Electrophysiology

Amphotericin B perforated configuration of patch clamp technique was used to record $K_V7$ current using a HEKA EPC10 amplifier and Patchmaster V2 (HEKA instruments). Voltage dependence of activation was investigated using a standard IV protocol consisting of a train of square voltage pulses (1000 ms) from –80 mV to +20 mV in 10 mV increments with a 600 ms deactivating pulse back to –80 mV; pulses were applied with 2 s interval. In this protocol, the tail current amplitude elicited by the deactivating step was measured to remove the impact of driving force on the recording. Pipettes were pulled using a horizontal puller (Sutter P-97) and fire polished to typically 2–4 MΩ resistance.

Upon entering whole-cell configuration, cell capacitance was nulled, and series resistance was typically below 10 MΩ.

Intracellular solution contained: 0.4 mg/mL Amphotericin B, 160 mM KCl, 5 mM MgCl2, 5 mM HEPES; pH adjusted to 7.4 using NaOH (all from Sigma).

Extracellular solution contained: 160 mM NaCl, 2.5 mM KCl, 1 mM $MgCl_2$, 10 mM HEPES, 2 mM $CaCl_2$, and 10 mM Glucose; pH adjusted to 7.4 with NaOH (all from Sigma).

## Expression and purification of the CaM/mTFP-K$_V$7.2-hAB-Venus complex and CaM-Venus

The K$_V$7.2-hAB segment (residues 316–532), K$_V$7.1-hAB (residues 250–412), K$_V$7.3-hAB (residues 349–556), K$_V$7.4-hAB (residues 316–571), and K$_V$7.5-hAB (residues 309–524) in complex with CaM and CaM-Venus recombinant protein were purified as previously described (*Bernardo-Seisdedos et al., 2018*).

## Mutant CaM

CaM3, CaM4, CaM123, CaM124, and CaM1234 in pOKD4 vector were synthesized by GenScript Biotech Corporation (Netherlands). CaM3 has mutation in the EF3 (D95A), CaM4 in the EF4 (D131A), CaM123 has mutations in EF1, EF2, and EF3 (D22A, D58A, and D95A), CaM124 has three alanine substitutions in the first, second, and fourth EF hands (D22A, D58A, and D131A), while CaM 1234 has substituted all EF hands (D22A, D58A, D95A, and D131A).

## Expression and purification of CaM

Recombinant CaM was produced in BL21 DE3 bacteria and purified as previously described (*Yus-Najera et al., 2002*).

## Peptide design and solubilization

Proteogenix prepared and purified by reverse phase HPLC (High performance liquid chromatography) chromatography all peptides, with greater than 95% purity (*Supplementary file 1*) and stored at –20°C lyophilized. Peptides were solubilized first in DMSO to get 10 mM concentration. To obtain 100 μM K$_V$7.2-S2/S3 solution, they were diluted in Fluorescence Buffer (Hepes 50 mM [pH 7.4], KCl 120 mM, NaCl 5 mM, $MgCl_2$ 2 mM, and EGTA 5 mM) with 1 mM DTT.

## Reversible peptide oxidation by H$_2$O$_2$

A 30% solution of $H_2O_2$ was from Thermo Fisher Scientific. The DMSO-peptide stock at 10 mM was diluted in oxidizing buffer (120 mM KCl, Hepes 50 mM pH 7.4, 5 mM EGTA, 5 mM NaCl, and 1 mM $H_2O_2$ [0.003%]). The oxidized peptide was added to samples with 500 nM FRET-sensor, both KCNQ2 and KCNQ4, in the absence and in the presence of 1 mM $CaCl_2$.

To reverse the oxidation, DTT was added at 1 mM to the previously oxidized peptide stock and incubated at 4°C O/N. Samples were prepared exactly the same, and FRET was analyzed as previously described.

## FRET experiments

All FRET experiments were carried using an AMINCON Bowman series 2 luminescence fluorimeter. The fluorescence emission spectra of the proteins at 500 nM were collected over 470–570 nM after excitation at 458 nm (2 nm bandwidth). The total protein in each condition was assessed by direct excitation of the yellow protein at 520 nm, collecting the emission at 520–570 nm. FRET index was established as the ratio of emission at 520–525 divided by emission at 485–490 nm upon excitation at 456–460 nm.

## Fluorometric measurements using D-CaM

Fluorescent D-CaM (5-[dimethylamino]naphtalene-1-sulfonyl-calmodulin) was prepared using recombinant CaM and dansyl chloride. Prior to the experiments, D-CaM and other proteins were dialyzed for 48 hr against 2 L of fluorescence buffer containing Hepes 50 mM (pH 7.4), KCl 120 mM, NaCl 5 mM, $MgCl_2$ 2 mM, and EGTA 5 mM, changing the buffer every 12 hr. Steady-state fluorescence measurements were obtained at 25°C on a SLM SPF-8100 (Olis, inc) fluorescence spectrophotometer in a final

volume of 100 μL. The fluorescence was measured after exciting at 340 nm and recording emission from 450 to 600 nm. Slit widths were set at 2 nm for excitation and 2 nm for emission.

Titration experiments were performed by adding increasing concentrations S2/S3 peptides to D-CaM (50 nM) in fluorescence buffer. Experiments were also performed in the presence of an excess of $Ca^{2+}$ by adding 4.34 mM $Ca^{2+}$ to the fluorescence buffer. The free $Ca^{2+}$ concentration was determined using Fura-2 (Invitrogen), following the manufacturer's instructions.

Fluorescence enhancement was plotted against the peptide concentration to generate the concentration-response curves. The parameters of the Hill equation were fitted to the data by curvilinear regression, enabling the apparent affinity ($EC_{50}$ or concentration that gives half-maximal change in the intensity of the fluorescence emission) or the $t_{50}$ (half-time or time that gives half-maximal change in the intensity of fluorescence emission). The data are shown as the average of three or more independent experiments.

## NMR measurements

Uniformly $^{15}N$-labeled $K_V7.2hAB:CaM$ was prepared in M9 medium containing 1 g/L $^{15}NH_4Cl$ as source of nitrogen. Other steps involved in the expression and purification of $^{15}N$-labeled $K_V7$-2hAB:CaM were the same as for mTFP-$K_V7.2hAB$-Venus:CaM. The $^2D$-$^1H,^{15}N$-HSQC experiments were performed by dissolving 75 μM complex in a buffer consisting of 50 mM Tris-HCl, 100 mM KCl, 10% $D_2O$, and 10 mM EGTA (apo-Complex) or, alternatively, 5 mM $CaCl_2$ (holo-Complex). The pH values of the sample solutions were carefully adjusted to 7.4 with trace amount of 2 M KOH. All NMR experiments were carried out at 25°C on a Bruker Avance III 600 MHz NMRs spectrometer. The HSQC (Heteronuclear single quantum correlation) spectra were acquired with a spectral width of 30 ppm in the $^{15}N$ dimension and 16 ppm in the $^1H$ dimension. In CSP experiments, peptide-CaM complexes were prepared by adding 13 equivalents of the $K_V7.1$-S2S3 lyophilized peptide directly to the Q2-$h$AB-CaM NMR sample. The CSP studies were performed by monitoring the changes in the $^1H,^{15}N$ HSQC spectra of $^{15}N$-labeled CaM. The CSP values were then evaluated as a weighted average chemical shift difference of $^1H$ and $^{15}N$ resonances, using the equation:

$$\delta = \sqrt{\left(0.1 * \delta_N\right)2 + \delta_{H^2}}.$$

## MDs simulations

Two systems were set up: the intermediate- (Int-CaM, N-lobe loaded with $Ca^{2+}$) and the holo-CaM/$K_V7.2$ hAB complexes (both lobes loaded with $Ca^{2+}$) in the presence of the $K_V7.1$ S2S3 peptide (164-RLWSAGCRSKYVGVWGRLRFARKP-187; PDB ID: 5VMS; *Sun and MacKinnon, 2017*). CaM/hAB complexes were modeled starting from the NMR structures of the $K_V7.2hAB/CaM$ complex (PDB IDs: 6FEG and 6FEH, for int- and holo-complexes, respectively; *Bernardo-Seisdedos et al., 2018*). The initial location of the peptide was established after aligning the EF3 lobes of the $K_V7.1/CaM$ and $K_V7.2/CaM$ complexes. Hydrogen atoms were added to the complexes using the VMD software (*Humphrey et al., 1996*). Two or four $Ca^{2+}$ atoms were added to the int- and holo-systems, located as in PDB 6FEG and 6FEH, respectively. The systems were then solvated resulting in a cubic box of dimensions described in *Supplementary file 2*. $K^+$ and $Cl^+$ ions were added using the Autoionize plugin of VMD to neutralize the system resulting in a final concentration of 150 mM KCl. The S2S3 peptide was also considered alone in solution to investigate its structural stability. Several replicas were run for some of the systems (*Supplementary file 2*).

Simulations were performed with NAMD2.13 (*Phillips et al., 2005*). The CHARMM36 force field (*Klauda et al., 2010*) was used to model the protein and ions, and the TIP3P model was chosen for the water. NBFIX corrections for $Ca^{2+}$ ions were adopted (*Yoo et al., 2016*). The Particle Mesh Ewald method was used for the treatment of electrostatic interactions, with an upper threshold of 1 Å for grid spacing (*Darden and Pedersen, 1993*). Electrostatic and van der Waals forces were calculated in every time step with a 12 Å cut-off distance. A switching distance of 10 Å was chosen to smoothly truncate the non-bonded interactions. Only atoms in a Verlet pair list with a 13.5 Å cut-off distance (reassigned every 20 steps) were considered. The SETTLE algorithm was used to constrain all bonds involving hydrogen atoms to allow the use of a 2 fs time step. The Nose-Hoover-Langevin piston method was employed to control the pressure with a 50 fs period, 25 fs damping constant,

and a desired value of 1 atmosphere. The system was coupled to a Langevin thermostat to sustain a temperature of 298°K throughout.

The VMD software was used for analysis of the trajectories and production of some of the figures. All replicas for each system were merged, and contacts of the Cα of all the residues were computed in the analysis of the contacts between S2S3 and CaM. The contacts were then normalized to the total time and to the value of the residue with maximum contacts.

## Acknowledgements

CM was supported by the Basque Government through a Basque Excellence Research Center (BERC) and JU was partially supported by BERC funds. CD thanks PRACE for awarding access to computational resources in CSCS, the Swiss National Supercomputing Service, in the 17th and 20th Project Access Calls. We acknowledge CESGA and CSIC for granting us access to computational resources to FinisTerrae II supercomputer.

## Additional information

### Funding

| Funder | Grant reference number | Author |
|---|---|---|
| Ministerio de Ciencia e Innovación | PID2021-128286NB-100 | Alvaro Villarroel |
| Wellcome Trust | 212302/Z/18/Z | Nikita Gamper |
| Medical Research Centre | MR/P015727/1 | Frederick Jones |
| Eusko Jaurlaritza | IT1707-22 | Alvaro Villarroel |
| Ekonomiaren Garapen eta Lehiakortasun Saila, Eusko Jaurlaritza | BG2019 | Alvaro Villarroel |
| Ministerio de Ciencia e Innovación | RTI2018-097839-B-100 | Alvaro Villarroel |
| Ministerio de Ciencia e Innovación | RTI2018-101269-B-I00 | Oscar Millet |
| Eusko Jaurlaritza | IT1165-19 | Alvaro Villarroel |
| Ekonomiaren Garapen eta Lehiakortasun Saila, Eusko Jaurlaritza | KK-2020/00110 | Alvaro Villarroel |
| Eusko Jaurlaritza | PRE_2018-2_0082 | Eider Nuñez |
| Eusko Jaurlaritza | POS_2021_1_0017 | Eider Nuñez |
| Eusko Jaurlaritza | PRE_2018-2_0126 | Arantza Muguruza-Montero |

The funders had no role in study design, data collection and interpretation, or the decision to submit the work for publication. For the purpose of Open Access, the authors have applied a CC BY public copyright license to any Author Accepted Manuscript version arising from this submission.

### Author contributions

Eider Nuñez, Formal analysis, Investigation, Methodology, Writing - review and editing; Frederick Jones, Arantza Muguruza-Montero, Ganeko Bernardo-Seisdedos, Oscar Millet, Data curation, Formal analysis, Investigation, Methodology, Writing - review and editing; Janire Urrutia, Supervision, Writing - review and editing; Alejandra Aguado, Supervision, Investigation, Methodology, Writing - review and editing; Covadonga Malo, Investigation, Methodology, Writing - review and editing; Carmen Domene, Data curation, Supervision, Methodology, Writing - review and editing; Nikita Gamper, Data curation, Supervision, Funding acquisition, Methodology, Writing - original draft, Writing - review and

editing; Alvaro Villarroel, Conceptualization, Formal analysis, Supervision, Funding acquisition, Methodology, Writing - original draft, Project administration, Writing - review and editing

### Author ORCIDs
Janire Urrutia  http://orcid.org/0000-0002-8546-292X
Nikita Gamper  http://orcid.org/0000-0001-5806-0207
Alvaro Villarroel  http://orcid.org/0000-0003-1096-7824

### Decision letter and Author response
Decision letter https://doi.org/10.7554/eLife.81961.sa1
Author response https://doi.org/10.7554/eLife.81961.sa2

---

## Additional files

### Supplementary files
- MDAR checklist
- Supplementary file 1. S2S3 petides information.
- Supplementary file 2. Details of the molecular dynamics simulations.

### Data availability
All data generated or analysed during this study are included in the manuscript and supporting file.

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
