## [Editor Report]

This useful study provides insights into mechanisms underlying oxidation regulation of Kv7 channels that contributes to regulating neuronal excitability. The experimental evidence in support of the major claims is solid, although it could be improved upon by studies on the holo-channel. The work will be of general interest to ion channel biophysicists and cell biologists.

---

## [Decision Letter]

**Decision letter after peer review:**

Thank you for submitting your article "Redox regulation of K_v_7 channels through EF3 hand of calmodulin" for consideration by *eLife*. Your article has been reviewed by 3 peer reviewers, and the evaluation has been overseen by a Reviewing Editor and Richard Aldrich as the Senior Editor. The following individual involved in the review of your submission has agreed to reveal their identity: Geoffrey S Pitt (Reviewer #2).

Essential revisions:

1) The model presented in Figure 7 is uninformative in its current form and needs to be redesigned and/or rewritten to more accurately convey information about the authors' findings. It is also important to include the caveat in any proposed model that conclusions are based on the use of isolated peptides in structural biochemical experiments that may not translate linearly to the whole channel complex.

2) There are apparent discrepancies or discordances in the data that need explanation.

Reviewer 3: In Figure 1A, endogenous CaM is sufficient to produce a response to H_2_O_2_ on par with CaM WT. However, in B endogenous CaM has a much larger basal current density, nearly identical to that of CaM3, CaM34, and CaM1234. By the proposed mechanism, this enhanced current density would be due to a loss of S2S3/CaM EF3 interaction, yet this would contradict the result in A. How do the authors then explain the loss of current density at baseline through the addition of CaM?

Reviewers 2 and 3: In Figure 3, the addition of the S2S3 peptide increases and shifts fluorescence, presumably through binding to apoCaM. The addition of Ca^2+^ also increases and shifts fluorescence. These two processes are described as additive. But this doesn't seem to match the overall mechanism where the authors claim Ca^2+^ binding to EF3 disrupts the binding between CaM and the S2S3 linker.

3) The S2S3 and CRDs from different members of the Kv7 family are used interchangeably in the study without appropriate justification. This is problematic because Kv7.1 and Kv7.3, for example, have been shown to be not regulated by oxidation. The rationale for using KCNQ1 S2S3 and the interpretation of results is not justified considering that KCNQ1 S2S3 has fewer Cys residues and was not affected by oxidation. Further, given that the functional data in Figure 1 were obtained for the Kv7.4 channel, it would make the most sense to present experiments using the S2S3 and CRD from Kv7.4 in the main results and move other isoforms to the supplemental data. Some discussion is also warranted as to why some Kv7 family members are not regulated by oxidation if the interactions among S2S3, CaM3, and the CRD are conserved among them all. Moreover, given this circumstance, the final sentence in the abstract – "Our data is consistent with the proposal that oxidation of cysteine residues in the S2S3 loop relieves Kv7 channels from a constitutive inhibition imposed by interactions between the EF3 hand of CaM which is necessary and sufficient for this signaling" – is an overstatement.

4) Presentation of patch clamp experiments and FRET experiments need to be made clearer. As indicated by reviewer 3, there are aspects of the presentation of these experiments that are either not provided or are poorly described which make it confusing to the reader.

*Reviewer #1 (Recommendations for the authors):*

1) Figure 6, more control experiments are needed: A) different S2S3 concentrations should be used as in Figure 5; B) H_2_O_2_ and DDT in the absence of S2S3 should be studied.

*Reviewer #2 (Recommendations for the authors):*

The one aspect of the paper that could be improved is the Discussion Figure (Figure 7). While the goal is to provide a simplified model, I don't find this figure to really "say" anything. Perhaps it is too simplified. The C-terminus (and the helixA-helixB fork) is missing from the overall channel diagram: providing it in the circle does not offer a clear model of what the authors likely wish to communicate. Further, the experiments (other than the electrophysiology) are focused on isolated peptides of the helixA-helixB fork and the S2-S3 peptide. How this influences the voltage sensor and its connection to the pore is too speculative. At the least, I would recommend a significant re-write/re-design of the model in Figure 7.

*Reviewer #3 (Recommendations for the authors):*

1. The patch clamp experiments are not well explained and seem to have errors in their description. For Figure 1, the authors explicitly state that they are measuring the 'outward steady-state current at -60 mV'. This is confusing as the protocol steps to -60mV from a holding potential of -20 mV, thus the channels will deactivate at -60. More confusing is the fact that the supplementary figures display the same protocol starting at a holding potential of -20mV. This appears to be incorrect based on the exemplars, which actually match the description in the methods which indicates steps from a holding potential of -80 mV. This lack of clear description of the experimental protocols used needs to be addressed.

2. The authors test the impact of CaM3 and CaM34 on channel function (Figure 1). Yet they do not probe CaM4. Given the nearly identical results for CaM3 and CaM34, it would be worth evaluating CaM4.

3. Figure 1, why does XE-991 not block the current in all conditions?

4. The functional data (Figure 1) was obtained for the KV7.4 channel. Subsequent studies utilized KV7.2. It is not clear to me why the authors made this switch as there is no mention of it in the text. This is critical as not all KV7 channels exhibit ROX-mediated regulation, thus it is not clear that the mechanism is entirely conserved across the channel subtypes.

5. In Figure 1A, endogenous CaM is sufficient to produce a response to H_2_O_2_ on par with CaM WT. However, in B endogenous CaM has a much larger basal current density, nearly identical to that of CaM3, CaM34, and CaM1234. By the proposed mechanism, this enhanced current density would be due to a loss of S2S3/CaM EF3 interaction, yet this would contradict the result in A. How do the authors then explain the loss of current density at baseline through the addition of CaM?

6. In Figure 3, the addition of the S2S3 peptide increases and shifts fluorescence, presumably through binding to apoCaM. The addition of Ca^2+^ also increases and shifts fluorescence. These two processes are described as additive. But this doesn't seem to match the overall mechanism where the authors claim Ca^2+^ binding to EF3 disrupts the binding between CaM and the S2S3 linker.

7. Please describe the experimental conditions for the FRET experiments in Figures2 and 5. What Ca^2+^ concentration is used in Figure 2? What CaM concentration is used in Figure 5? It would be helpful to have some idea of the molar ratio between CaM and the peptide considering the reduction in FRET that seems to occur at higher concentrations.

8. In Figure 5, Ca^2+^ binding still alters FRET in presence of S2S3 with CaM3 – doesn't this by itself show that EF3 is not acting entirely alone making the assertion of necessary and sufficient inaccurate? At most, the authors should say it is an important element, but they have not proven the full statement.

9. Figure 5 is impossible to interpret as presented. By normalizing the data to zero Ca^2+^, I cannot actually tell what adding the S2S3 peptide did in each case – please put this data on the same scale as Figure 2. Since the data is reported as FRET index, a value which by definition cannot cross zero, the two directions of data are confusing. I'm finding it impossible to correlate the data in Figure 2 with that in Figure 5. In particular, a Ca^2+^ dependent reduction in the FRET index as described by the authors for Figure 5 is confusing when looking at Figure 2 – where the FRET index increases in the presence of preserved Ca^2+^ binding of EF3. Either this is a change in the display metric, or there is a discrepancy between the data and the conclusions being drawn.

---

## [Author Response]

Essential revisions:1) The model presented in Figure 7 is uninformative in its current form and needs to be redesigned and/or rewritten to more accurately convey information about the authors' findings. It is also important to include the caveat in any proposed model that conclusions are based on the use of isolated peptides in structural biochemical experiments that may not translate linearly to the whole channel complex.

1A. We have included, as suggested, the caveat that the conclusions derived from isolated peptides may not translate linearly to the whole channel complex (lines 424-425):

“However, conclusions derived from the use of isolated peptides may not translate linearly to the whole channel complex.”

1B. We agree that the model is speculative and that it does not clearly convey information about our finding. Consequently, we have removed Figure 7.

2) There are apparent discrepancies or discordances in the data that need explanation.Reviewer 3: In Figure 1A, endogenous CaM is sufficient to produce a response to H_2_O_2_ on par with CaM WT. However, in B endogenous CaM has a much larger basal current density, nearly identical to that of CaM3, CaM34, and CaM1234. By the proposed mechanism, this enhanced current density would be due to a loss of S2S3/CaM EF3 interaction, yet this would contradict the result in A. How do the authors then explain the loss of current density at baseline through the addition of CaM?

2.1 The observed changes in current density after over-expression of CaM reflect changes in the number of channels at the membrane, the channel activity, or both. Over-expression of CaM is known to influence trafficking of Kv7 channels, therefore affecting the number of channels at the plasma membrane.

The data on Figure 1B shows that current density in larger in cells over-expressing CaM variants in which EF3 hand is mutated (CaM3, CaM34, CaM1234) compared to those cells over-expressing variants with the EF3 not mutated (WT, CaM12, -the results with the CaM124 are intermediate-). These results are compatible with WT CaM causing a tonic inhibition of the channel mainly through EF3. Yet, the interpretation requires caution. We have added the following sentence (lines 174-176):

“Yet, the interpretation of this effect requires caution, since CaM over-expression also affects the number of the channels at the plasma membrane (38, 39).”

We referred to this issue in the original manuscript (lines 412-414):

To harmonize with other observations, we propose that this inhibition is counterbalanced by CaM-dependent promotion of surface expression when CaM or CaM1234 are over-expressed (38, 39).

Reviewers 2 and 3: In Figure 3, the addition of the S2S3 peptide increases and shifts fluorescence, presumably through binding to apoCaM. The addition of Ca^2+^ also increases and shifts fluorescence. These two processes are described as additive.

2.2 We found that Ca^2+^ causes a dose-dependent leftward shift in the peak of fluorescence and an increase in the fluorescence intensity. There is no further (or it is negligible) increase in leftward shift and intensity at higher Ca^2+^ concentrations. Addition of the S2S3 peptide in the presence of high Ca^2+^ causes a further shift and an increase in the peak emission intensity.

But this doesn't seem to match the overall mechanism where the authors claim Ca^2+^ binding to EF3 disrupts the binding between CaM and the S2S3 linker.

2.3 We do not suggest that Ca^2+^ binding to EF3 DISRUPTS the binding between CaM and the S2S3 linker, we suggest that Ca^2+^ binding to EF3 weakens the interaction between CaM and the S2S3 linker. Based on the data on Figure 3, we suggest that “Ca^2+^ mitigates the effect of the peptide on D-CaM” (line 246).

We think that the interaction of the S2S3 is COMPATIBLE with Ca^2+^ binding. In other words, we propose that binding to S2S3 is less favorable when CaM is interacting with Ca^2+^. Figure 3 and Figure 3—figure supplement 2 shows that a higher S2S3 peptide concentration is required to cause half of the maximal effect on D-CaM fluorescence increase in the presence of saturating free Ca^2+^ concentration compared to the nominal absence of Ca^2+^.

We state that “Ca^2+^ and the peptide can interact with CaM SIMULTANEOUSLY” (line 245) which fits nicely with the suggestion that the redox effect requires the presence of Ca^2+^ (Figure 1). We also stated (lines 303-304) “S2S3 is not canceling the effect of Ca^2+^ by competing or displacing this cation from its binding site”. We further suggested (lines 308-310): “Thus, the direction/orientation of the movements in the AB fork when EF3 is loaded with Ca^2+^ is reversed upon interaction with S2S3”.

3) The S2S3 and CRDs from different members of the Kv7 family are used interchangeably in the study without appropriate justification.

3.1 The results from different S2S3 peptides and from different CRDs are very similar (Figure 3 and Figure 3-suppl 2, Figure 5 and Figure 5-Suppl 1). This is an important finding, and suggests that the underlying mechanism is present in all members of the Kv7 channel family.

This is problematic because Kv7.1 and Kv7.3, for example, have been shown to be not regulated by oxidation.

3.2 We have addressed this issue in the discussion (lines 371-387):

“Earlier, we showed that the redox response depends on the presence of cysteine residues at the S2S3 loop (10); yet, the number of cysteine residues is not the sole factor defining the efficacy of the response. No evidence for redox regulation was observed for WT K_V_7.1 channels (that have only one cysteine residue at the S2S3 loop); and engineered “three Cys K_V_7.1 channel” displayed a weak response to H_2_O_2_. WT K_V_7.4 (three Cys residues in the S2S3) displayed a strong response to H_2_O_2_, yet a partial response was still observed for engineered “one Cys K_V_7.4 channels” (10). Hence, even with only one cysteine present, the S2S3 linker can mediate H_2_O_2_ sensitivity of a ‘responsive’ K_V_7 channel (such as K_V_7.4), while there must be other structural constrains that hinder potentiation of K_V_7.1 by the H_2_O_2_.

We suggest that a low channel open probability (p_o_) is an additional requirement for redox sensitivity, because no evidence for redox regulation has been observed for K_V_7.3 channels, which present a p_o_ close to the unity at saturating voltages (47). These channels have a triple Cys residue motive at the S2S3 loop that only differs at one position when compared to redox-sensitive K_V_7.2 channels (an Arg residue in K_V_7.3 versus a Lys residue in K_V_7.2). We suggest that due to very little room for further increase in p_o_, oxidation has negligible consequences on macroscopic K_V_7.3 currents.”

The rationale for using KCNQ1 S2S3 and the interpretation of results is not justified considering that KCNQ1 S2S3 has fewer Cys residues and was not affected by oxidation.

3.3 We agree with the reviewer comments. At the onset of this project only the K_V_7.1 structure was available and only the FRET K_V_7.2/CaM biosensor complex was characterized. This has driven the original strategy. To address this concern we have now replaced the indicated datasets with new data obtained with K_V_7.4 and K_V_7.2 peptides and biosensors. As mentioned above, a partial response to H_2_O_2_ was originally observed for engineered “one Cys K_V_7.4 channels”. Hence, even the S2S3 linker with one cysteine present appears sufficient to mediate H_2_O_2_ sensitivity of a ‘responsive’ K_V_7 channel (such as K_V_7.4), while there must be other structural constrains that hinder potentiation of K_V_7.1 by the H_2_O_2_.

Further, given that the functional data in Figure 1 were obtained for the Kv7.4 channel, it would make the most sense to present experiments using the S2S3 and CRD from Kv7.4 in the main results and move other isoforms to the supplemental data.

3.4 We have exchanged the panel corresponding to K_V_7.4 from Supplementary Figure 5 to Figure 3.

To address these concerns, we have performed additional experiments with peptides with the K_V_7.2 and K_V_7.4 sequences (three cysteine residues). These new data are presented in Figure 6. The previous data has been included as Figure 6-Figure suppl 2.

Some discussion is also warranted as to why some Kv7 family members are not regulated by oxidation if the interactions among S2S3, CaM3, and the CRD are conserved among them all.

3.5 See response in 3.2.

Moreover, given this circumstance, the final sentence in the abstract – "Our data is consistent with the proposal that oxidation of cysteine residues in the S2S3 loop relieves Kv7 channels from a constitutive inhibition imposed by interactions between the EF3 hand of CaM which is necessary and sufficient for this signaling" – is an overstatement.

3.6 We have removed that sentence.

4) Presentation of patch clamp experiments and FRET experiments need to be made clearer. As indicated by reviewer 3, there are aspects of the presentation of these experiments that are either not provided or are poorly described which make it confusing to the reader.

4.1 We have simplified the presentation of FRET data of Figure 5.

Reviewer #1 (Recommendations for the authors):1) Figure 6, more control experiments are needed: A) different S2S3 concentrations should be used as in Figure 5; B) H_2_O_2_ and DDT in the absence of S2S3 should be studied.

R1.3 We have performed new experiments, and the new data is consistent with our original conclusions. A new Figure 6 has been included and the results of the requested control experiments have been included in Supplementary Figure 12.

Reviewer #2 (Recommendations for the authors):The one aspect of the paper that could be improved is the Discussion Figure (Figure 7). While the goal is to provide a simplified model, I don't find this figure to really "say" anything. Perhaps it is too simplified. The C-terminus (and the helixA-helixB fork) is missing from the overall channel diagram: providing it in the circle does not offer a clear model of what the authors likely wish to communicate. Further, the experiments (other than the electrophysiology) are focused on isolated peptides of the helixA-helixB fork and the S2-S3 peptide. How this influences the voltage sensor and its connection to the pore is too speculative. At the least, I would recommend a significant re-write/re-design of the model in Figure 7.

R2.1 We agree with the referee that the model of Figure 7 is speculative, and we have removed this Figure.

Reviewer #3 (Recommendations for the authors):1. The patch clamp experiments are not well explained and seem to have errors in their description. For Figure 1, the authors explicitly state that they are measuring the 'outward steady-state current at -60 mV'. This is confusing as the protocol steps to -60mV from a holding potential of -20 mV, thus the channels will deactivate at -60. More confusing is the fact that the supplementary figures display the same protocol starting at a holding potential of -20mV. This appears to be incorrect based on the exemplars, which actually match the description in the methods which indicates steps from a holding potential of -80 mV. This lack of clear description of the experimental protocols used needs to be addressed.

R3.1 We believe that there are no mistakes in the figure, as explained next. We used two different voltage protocols, one for monitoring of steady-state current activity and another one for studying voltage-dependence. In the first instance, a square voltage pulses from -20 to -60 mV are applied every 2 seconds. Kv7 channels have very negative activation threshold (and they do not inactivate), so even at -60 mV a fraction of channels remains open. Hence, we monitored the residual outward current at -60 mV, as in the cartoon below:

**Author response image 1. sa2fig1:** 

The advantage of this method is in that it minimizes contamination of recording with endogenous K^+^ channels, as most of the other voltage-gated K^+^ channels activate at more positive voltages and/or inactivate.On the other hands, the voltage protocol for current-voltage relationships (Suppl. Figure 1, Suppl. Figure 2C,D) starts from a holding potential of -80 mV and goes in steps from -100 to +50 mV with tail current amplitudes being plotted against voltage.

A clarification has been added to the relevant sections of the manuscript (lines 136-139).

“Cells were held at -20 mV and 600 ms voltage pulses to -60 mV were applied every 2 s; Kv7.4 activity was monitored as the outward steady-state current amplitude at -60 mV (Figure 1C).”

2. The authors test the impact of CaM3 and CaM34 on channel function (Figure 1). Yet they do not probe CaM4. Given the nearly identical results for CaM3 and CaM34, it would be worth evaluating CaM4.

R3.2 We actually tested the effects of CaM12, CaM124, CaM3, CaM34 and CaM1234. Given that CaM12 and CaM124 behave similar to WT CaM while CaM3 and CaM34 behave similar to CaM1234 (DN CaM), the conclusion of the negligible contribution of hand 4 seems pretty sound. It is true that other combinations could have been tested as well, but we think that the range of combination that we have tested provides a very conclusive outcome.

3. Figure 1, why does XE-991 not block the current in all conditions?

R3.3 XE991 at 10 µM blocked Kv7 currents in all recordings. In some instances the block was somewhat incomplete. It is important to keep in mind that XE991 has an IC_50_ for Kv7 channels in the range of 1 µM, hence, at 10 µM some residual current is not unexpected. We generally refrain from using higher concentrations of XE991 to avoid non-specific effects.

4. The functional data (Figure 1) was obtained for the KV7.4 channel. Subsequent studies utilized KV7.2. It is not clear to me why the authors made this switch as there is no mention of it in the text. This is critical as not all KV7 channels exhibit ROX-mediated regulation, thus it is not clear that the mechanism is entirely conserved across the channel subtypes.

R3.4 We addressed the issue of differential sensitivity of Kv7 channels to H_2_O_2_ in the sections 3.2 and R1.1 above (and in the discussion, lines 371-387).

We have performed new experiments with Kv7.2 and Kv7.4 peptides (3 cysteine residues). These new data confirm our conclusions, and are now included in Figure 6.

5. In Figure 1A, endogenous CaM is sufficient to produce a response to H_2_O_2_ on par with CaM WT. However, in B endogenous CaM has a much larger basal current density, nearly identical to that of CaM3, CaM34, and CaM1234. By the proposed mechanism, this enhanced current density would be due to a loss of S2S3/CaM EF3 interaction, yet this would contradict the result in A. How do the authors then explain the loss of current density at baseline through the addition of CaM?

R3.5 We address this question in section 2.1.

6. In Figure 3, the addition of the S2S3 peptide increases and shifts fluorescence, presumably through binding to apoCaM. The addition of Ca^2+^ also increases and shifts fluorescence. These two processes are described as additive. But this doesn't seem to match the overall mechanism where the authors claim Ca^2+^ binding to EF3 disrupts the binding between CaM and the S2S3 linker.

R3.6 We address this question in section 2.3.

7. Please describe the experimental conditions for the FRET experiments in Figures2 and 5. What Ca^2+^ concentration is used in Figure 2? What CaM concentration is used in Figure 5? It would be helpful to have some idea of the molar ratio between CaM and the peptide considering the reduction in FRET that seems to occur at higher concentrations.

R3.7 The requested information is included in the Figure legends now.

8. In Figure 5, Ca^2+^ binding still alters FRET in presence of S2S3 with CaM3 – doesn't this by itself show that EF3 is not acting entirely alone making the assertion of necessary and sufficient inaccurate? At most, the authors should say it is an important element, but they have not proven the full statement.

R3.8 We agree with the referee, and we realize that our statement may be somehow misleading. We acknowledged that EF3 is not acting alone: “Thus, EF3 plays a significant role in transmitting Ca^2+^ signals to the AB fork, and EF4 plays a secondary function” (lines 219-221). To avoid confusions, we have replaced that statement for “critical”, “crucial” or “central”.

9. Figure 5 is impossible to interpret as presented. By normalizing the data to zero Ca^2+^, I cannot actually tell what adding the S2S3 peptide did in each case – please put this data on the same scale as Figure 2. Since the data is reported as FRET index, a value which by definition cannot cross zero, the two directions of data are confusing. I'm finding it impossible to correlate the data in Figure 2 with that in Figure 5. In particular, a Ca^2+^ dependent reduction in the FRET index as described by the authors for Figure 5 is confusing when looking at Figure 2 – where the FRET index increases in the presence of preserved Ca^2+^ binding of EF3. Either this is a change in the display metric, or there is a discrepancy between the data and the conclusions being drawn.

R3.9 We thank the referee for this suggestion. We have plotted the data in a similar fashion than Figure 2. To do so, we have removed the dose-response curves, which are now presented in Supplementary Figure 10.